# Adaptive Contracts for Cost-Effective AI Delegation

**Eden Saig** [1 2]   **Tamar Garbuz** [* 3]   **Ariel D. Procaccia** [* 4]   **Inbal Talgam-Cohen** [* 3 1]   **Jamie Tucker-Foltz** [* 4 5]

## Abstract

When organizations delegate text generation tasks to AI providers via pay-for-performance contracts, expected payments rise when evaluation is noisy. As evaluation methods become more elaborate, the economic benefits of decreased noise are often overshadowed by increased evaluation costs. In this work, we introduce *adaptive contracts* for AI delegation, which allow detailed evaluation to be performed selectively after observing an initial coarse signal in order to conserve resources. We make three sets of contributions: First, we provide efficient algorithms for computing optimal adaptive contracts under natural assumptions or when core problem dimensions are small, and prove hardness of approximation in the general unstructured case. We then formulate alternative models of randomized adaptive contracts and discuss their benefits and limitations. Finally, we empirically demonstrate the benefits of adaptivity over non-adaptive baselines using question-answering and code-generation datasets.

## 1. Introduction

The delegation of tasks to LLMs is becoming increasingly prevalent, including high-stake tasks such as summarizing medical records or modifying critical software systems. For such tasks, organizations typically desire the best (and most costly) model available. However, a salient issue is lack of transparency—models are typically offered as black-boxes operated by LLM service providers, who interally operate the computational infrastructure required for inference. This creates a market failure known as *moral hazard*.

Moral hazard incentivizes providers to cut costs by covertly using cheaper, lower-quality models behind the scenes. This phenomenon is also known as *model substitution* (Cai et al., 2025) or *quality downgrade* (Sun et al., 2025). It is tightly linked to current LLM monetization—as Sun et al. put it, while bills are visible to the user, the tokens being billed are not. Cai et al. investigate a host of methods for auditing model substitution, concluding that detection based solely on LLM responses is hard. They suggest more radical solutions, requiring the release of statistical data or the adoption of hardware-assisted verification.

Against this backdrop, enter *contract design*, a field of economics dedicated to mitigating moral hazard through better, *performance-based* pricing (Holmstrom & Milgrom, 1987). Our starting point is work showing that this tool is well-worth exploring in the context of LLMs (Saig et al., 2024; Velasco et al., 2025). Whether for organization with market power who can negotiate their own payment scheme (e.g., a large healthcare vendor), or for LLM service providers looking to improve their competitiveness by adopting more user-aligned pricing, the lens of contract design highlights new ways of monetizing LLMs while benefitting consumers.

Pay-for-performance pricing requires performance evaluation, which in our context is quality evaluation of the LLM-generated response. The rapidly-growing literature on LLM evaluation and LLM benchmarks indicates that evaluation is not easy and is certainly not without cost (Stureborg et al., 2024; Shi et al., 2024; Chen et al., 2024; Fish et al., 2025). Apart from direct costs, there is an indirect loss due to the inevitable noisiness of evaluation, which puts the LLM provider at risk of investing in a high-cost model, but receiving low performance evaluation. Due to this risk, the provider might under-invest, causing inefficiency.

Our goal in this work is to study the cost-benefit trade-off between *coarse* performance evaluation and its risks, and *refined but costly* evaluation on the other hand. To achieve this, we follow classic economic literature on monitoring, and extend the classic contract design framework to allow for adaptive acquisition of performance information. We formulate an *adaptive contract* setting: a simple setup in which there are two levels of evaluation, the second one refining the first but at additional cost. The decision whether to acquire the second evaluation is adaptive, i.e., may depend on the result of the first evaluation. The following two

---

[*] After Eden Saig, authors are listed in alphabetical order.
[1] Technion – Israel Institute of Technology [2] California Institute of Technology [3] Tel Aviv University [4] Harvard University [5] Yale School of Management. Correspondence to: Eden Saig <edens@caltech.edu>, Tamar Garbuz <garbuz@mail.tau.ac.il>, Ariel Procaccia <arielpro@gmail.com>, Inbal Talgam-Cohen <inbaltalgam@gmail.com>, Jamie Tucker-Foltz <j.tuckerfoltz@yale.edu>.

*Proceedings of the 43$^{rd}$ International Conference on Machine Learning*, Seoul, South Korea. PMLR 306, 2026. Copyright 2026 by the author(s).

examples illustrate scenarios captured by our setting:

**Example 1.1** (Costly secondary evaluation). Consider the task of question answering by an LLM, which can use increasingly sophisticated and costly models, say GPT-5 nano, GPT-5 mini, and GPT-5.2. The user cannot directly observe which model is being used. Some evaluation methods are costless but noisy, like checking the length of the output, or the existence of different keywords. Other methods such as LLM-as-a-judge or human annotation are more accurate, but expensive (e.g., Zheng et al., 2023; Li et al., 2023). In two-tiered evaluation, we may first check e.g. the response length, then decide whether to pay for a costly annotation.

**Example 1.2** (Adaptive randomized evaluation). Consider a large organization seeking to deploy a language model to fix open issues ('bugs') in a large codebase. Attempting to resolve all existing issues is typically infeasible at the time of transaction, and therefore quality is typically evaluated on a set of issues chosen at random. In two-tiered evaluation, we may conserve resources by running a small set of tests first, then optionally run a larger suite based on initial results.

Note that in Examples 1.1 and 1.2, the user needs to decide (i) whether to further inspect the initial signal to get a refined one, and (ii) how much to pay the provider based on performance evaluation(s). We call the combination of inspection and payment schemes an *adaptive contract*, and study how to optimize its design for the user.

**Contributions.** We characterize the computational complexity of finding optimal adaptive contracts. Such contracts exhibit unique economic properties that differentiate them from previously-studied contract classes: in particular, there exist settings where no such contract is able to extract *any* utility for the user, despite strictly-positive utility of the provider from the interaction.[1] Due to these properties, optimizing over this class of contracts poses new challenges. *As our positive result*, we show in Section 3 that the problem of adaptive contract optimization admits a polynomial-time solution in two key practical settings: (i) when either the number of available generative models or the number of evaluation results is constant (see Theorem 3.2), and (ii) when the evaluations are independent, as in Example 1.2 (see Theorem 3.5). *On the negative side*, in Section 4 we show that outside of the above two settings, the design of adaptive contracts becomes NP-hard, and the hardness extends even to approximation by any constant factor. In this sense, our positive results are tight: the problem is intractable beyond constant dimensions and independent evaluations (see Theorems 4.1 and 4.2).

In Section 5, we extend our class of adaptive contracts to allow for *randomized inspection*, where a coin-flip determines whether or not the user will acquire a refined evaluation. Intuitively, randomized inspection can lower costs for the user. This turns out to work unrealistically well, effectively driving inspection costs to zero: in theory, the user can continue to reduce expected inspection costs ad infinitum, by promising increasingly high payments contingent on the inspection, but inspecting with vanishing probabilities. In game theoretic terms, the issue is that the resulting game does not admit a Stackelberg equilibrium (see Theorem 5.1).

Motivated by this negative property, we propose two alternative restrictions on the user, which rule out the unrealistic capability to inspect at no cost, and restore equilibrium guarantees: (1) The user cannot contractually commit to inspection probabilities; indeed, such a commitment would be difficult to enforce in some scenarios. (2) The user cannot promise unbounded payments, since payments absent inspection upper-bound payments following inspection; indeed, in scenarios like Example 1.1, a closer inspection of the code is likely to reveal bugs, leading to payment reductions. We prove that each of these restrictions is sufficient to restore equilibrium existence (see Theorem 5.4), and that subject to either restriction, randomized inspection can only help the user—sometimes strictly so.

Our experiments in Section 6 show how adaptive contracts better help LLM users while still compensating LLM providers, as compared to current LLM pricing.

**Related work.** Our work contributes to the growing literature on algorithmic contract design—see Dütting et al. (2024) for a recent survey. Closest to our work is that of Dye (1986), which considers a similar model of costly inspection, showing that under a list of assumptions in a stylized, one-dimensional model, the optimal contract admits a very simple inspection policy. Two of these assumptions are quite natural and also considered in this paper, namely MLRP and ISOP (see Section 2). Other assumptions make less sense for our setting, such as strict risk-aversion and a monotonicity property between quality and inspection costs.

Also closely related are two papers which introduce contract design with *action* inspection (Fallah & Jordan, 2024; Ezra et al., 2026). Using the language of our work for easy comparison, action inspection means that the user can pay to learn the generative model chosen by the provider. In contrast, our setting allows *outcome* inspection. I.e., for additional payment, the user can obtain a better quality indicator. More broadly, incorporating adaptive information acquisition places our work within a recent line of research studying information and contract design Castiglioni & Chen (2025); Babichenko et al. (2024); Garrett et al. (2023). In another direction, (Neyman et al., 2021) study how scoring rules

---

[1] In standard (non-adaptive) contract design, a simple linear contract always guarantees positive utility; but in an adaptive setting, inspection may be necessary for non-zero utility, and its cost might cancel out the otherwise-guaranteed positive utility.

incentivize adaptive information acquisition—but by the forecaster for which they are designed, rather than by the designer of the contract as in our setting.

## 2. Adaptive Contract Setting

In this section, we present an economic setting that generalizes classic contract design to allow for refined performance evaluation, and formalize the connection to LLMs . Our setting is illustrated in Figure 1.

**Notation.** For an $n \times m$ matrix $\mathbf{q}$ and indices $i \in [n], j \in [m]$, let $\mathbf{q}_i$ be the $i$th row and let $q_{i,j}$ be the $j$th entry of $\mathbf{q}_i$.

**Setting.** The setting consists of two players, a *principal* and an *agent*—the user and provider in the LLM scenario. In alignment with contract theory conventions, the agent has a set of possible *actions* $[n]$ with increasing costs $0 \leq c_1 \leq \cdots \leq c_n$ (borne by the agent). In the LLM scenario, each action represents a generative model, and its associated cost reflects the computational resources required for inference. After the agent chooses an action $i \in [n]$, the principal receives an initial *signal* $k \in [\ell]$, drawn from a distribution $\mathbf{q}_i^0$. In the LLM scenario, the signal is the coarse evaluation.

Now diverging from standard contract settings, the principal adaptively decides whether to further inspect signal $k$ at cost $d_k \geq 0$ (borne by the principal). Inspection reveals an *outcome* $j \in [m_k]$ drawn from a distribution $\mathbf{q}_i^k$ (note that $\mathbf{q}_i^k$ depends on signal $k$; in Section 3 we consider the case in which the outcome $j$ and signal $k$ are independent). In the LLM scenario, the outcome is the costly evaluation. The signal-outcome pair $(k, j)$ determines the *reward* $r_{k,j}$ of the principal from the agent's action. In the LLM scenario, the reward is the value eventually generated for the user by the LLM. An adaptive contract setting is summarized by a tuple $(\mathbf{q}^0, \mathbf{q}^1, \ldots, \mathbf{q}^\ell, \mathbf{c}, \mathbf{d}, \mathbf{r})$, where $\mathbf{q}^0$ is an $n \times \ell$ matrix whose $i$th row is the signal distribution $\mathbf{q}_i^0$, and $\mathbf{q}^k$ for $k \in [\ell]$ is an $n \times m_k$ matrix whose $i$th row is the outcome distribution $\mathbf{q}_i^k$. Vector $\mathbf{c}$ contains the action costs, and vector $\mathbf{d}$ the signal inspection costs. An $\ell \times \overline{m}$ matrix $\mathbf{r}$ contains the rewards for every (signal, outcome) pair $(k, j)$, where $\overline{m} = \max_{k \in [\ell]} m_k$ (and $\perp$ if $j > m_k$).

**MLRP.** We adopt an assumption standard to contract settings (e.g., (Dütting et al., 2024)) called *Monotone Likelihood Ratio Property (MLRP)*. An $n \times m$ matrix $\mathbf{q}$ whose rows are distributions is said to satisfy MLRP if for every pair of rows $i < i'$, the likelihood ratio $q_{i',j'}/q_{i,j'}$ is increasing in $j' \in [m]$. In an adaptive contract setting, matrices $\mathbf{q}^0, \mathbf{q}^1, \ldots, \mathbf{q}^\ell$ are assumed to satisfy MLRP. In the LLM scenario, this means that higher evaluations are more likely given a high-cost generative model than a low-cost one. Theorem 4.2 shows how our results change in absence of MLRP.

**Adaptive contract.** Before the interaction begins, the principal commits to an adaptive contract $(\mathbf{p}, \mathbf{s}, \mathbf{t})$ consisting of

inspection and payment policies. Binary vector $\mathbf{p} \in \{0, 1\}^\ell$ indicates whether to inspect each signal $k \in [\ell]$ (in Section 5 we consider probabilistic inspections, i.e., $\mathbf{p} \in [0, 1]^\ell$). Vector $\mathbf{s} \geq 0$ contains the payments to the agent for signals that are not inspected. Finally, $t_{k,j} \geq 0$ is the payment to the agent for signal-outcome pair $(k, j)$. The assumption that all payments are non-negative is known as *limited liability* and is fundamental in contract settings (Dütting et al., 2024).

**Agent's strategic response.** Given an adaptive contract $(\mathbf{p}, \mathbf{s}, \mathbf{t})$, the expected payment from principal to agent for taking action $i$ is $T_i = T_i(\mathbf{p}, \mathbf{s}, \mathbf{t}) := \mathbb{E}_{k \sim \mathbf{q}_i^0}[(1 - p_k)s_k + p_k \mathbb{E}_{j \sim \mathbf{q}_i^k}[t_{k,j}]]$ (we omit $(\mathbf{p}, \mathbf{s}, \mathbf{t})$ from the notation where clear from the context). The agent's expected utility for taking action $i$ is $U_A(i) := T_i - c_i$, and thus the agent's *best response* is $i^* \in \arg\max_i U_A(i)$ (where following the standard convention, ties are broken in favor of the principal).

**Principal's goal.** The principal's expected reward when the agent takes action $i$ is $R_i := \mathbb{E}_{k \sim \mathbf{q}_i^0}[\mathbb{E}_{j \sim \mathbf{q}_i^k}[r_{k,j}]]$, and the expected inspection cost is $D_i := \mathbb{E}_{k \sim \mathbf{q}_i^0}[p_k d_k]$. For a given adaptive contract $(\mathbf{p}, \mathbf{s}, \mathbf{t})$, the principal's expected utility is $U_P(\mathbf{p}, \mathbf{s}, \mathbf{t}) = R_{i^*} - T_{i^*} - D_{i^*}$, where $i^*$ is the agent's best-response action. A *MIN-PAY* contract for action $i$ maximizes the principal's expected utility while incentivizing the agent to choose action $i$, by minimizing the total expected payment $T_i + D_i$ of the principal subject to $i^* = i$. Our aim is to design an *optimal* adaptive contract maximizing $U_P(\mathbf{p}, \mathbf{s}, \mathbf{t})$. Alternatively, we aim for an optimal contract targeting the highest-cost action (the MIN-PAY contract for action $n$).

## 3. Computing Optimal Adaptive Contracts

In this section we seek algorithms for computing an optimal adaptive contract. While adaptivity poses new challenges, we identify two natural families of settings for which optimization is tractable. Missing proofs appear in Appendix E.

**Challenge 1: Non-linear programming.** A common approach to computing optimal *classic* contracts uses mathematical programming: For every action $i \in [n]$, find a MIN-PAY contract for $i$ by solving a linear program; then pick the overall best action to incentivize. Computing an optimal *adaptive* contract can also be reduced to finding $n$ MIN-PAY contracts, but this time finding such a contract requires solving a mixed-integer QCQP (quadratically-constrained quadratic program). These are generally intractable and require full enumeration over policies $\mathbf{p} \in \{0, 1\}^\ell$ to find the best one, thus we defer their formulation to Appendix D.1. In what follows, a useful fact is that if we fix the inspection policy $\mathbf{p}$, then optimizing the payments $\mathbf{s}, \mathbf{t}$ once again reduces to a linear program solvable in polynomial time.

**Challenge 2: Inspection cost offsets value.** Consider the following example:

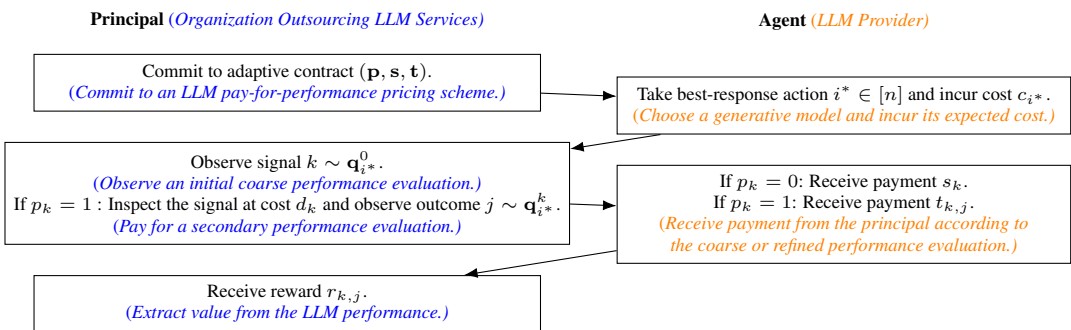

*Figure 1.* Principal-agent interaction in an abstract adaptive contract setting. Colored text describes the application to AI task delegation.

**Example 3.1.** Let $n = 2$ (two available actions), $\ell = 1$ (one coarse signal), $m_1 = 2$ (two inspection outcomes). Given these dimensions we define an adaptive contract setting $(\mathbf{q}^0, \mathbf{q}^1, \mathbf{c}, \mathbf{d}, \mathbf{r})$: For matrix $\mathbf{q}^0$, necessarily $\mathbf{q}_1^0 = \mathbf{q}_2^0 = (1)$, since the single signal is observed with probability $1$, regardless of the action. For matrix $\mathbf{q}^1$, let $\mathbf{q}_1^1 = (1, 0)$ and $\mathbf{q}_2^1 = (0, 1)$; this means that inspecting the signal reveals the action, by deterministically leading to outcome $1$ given action $1$, and to outcome $2$ given action $2$. Let the action costs be $\mathbf{c} = (0, 1)$, and the signal cost $d_1 = 1$. Finally, only the second outcome is rewarding: $\mathbf{r} = (0, 2)$.

To incentivize the first action, the MIN-PAY contract need not inspect the signal, and no payment is necessary. To incentivize the second action, the MIN-PAY contract must inspect the signal at cost $d_1 = 1$, and pay $t_{1,2} = 1$ for outcome $2$ and $0$ otherwise. In both cases, the principal's utility is $0$, so either contract is optimal. Interestingly, the principal-agent interaction yields no utility, despite the interaction's economic value if action $2$ is incentivized. Action $2$ leads to reward $2$ and costs only $1$; in economic terms, the *first best* (achievable through perfect cooperation) is $2 - 1 = 1$. The underlying reason for zero utility despite positive first best is that the initial signal is uninformative. Thus, action $2$ cannot be incentivized without further inspection, and the cost of inspection exactly offsets the first best. In comparison, in classic settings without inspection, a simple *linear* contract already guarantees the principal a fraction of the first best. This demonstrates that adaptive contract settings exhibit distinct economic properties, raising new challenges.

**Family 1: Settings with constant-many actions.** We show that the optimal adaptive contract problem is tractable with constant-many actions. (It is not hard to show a similar result for constant-many evaluation outcomes.)

**Theorem 3.2.** *Consider adaptive contract settings with constant-many actions. Then an optimal adaptive contract can be computed in polynomial time.*

Our proof relies on a combination of two insights: First, a sparsity argument by which if the number of actions is bounded by a constant, then there exists an optimal con-

tract where the number of inspected signals is bounded by the same constant, significantly reducing the search space. Furthermore, we show that if a contract inspects a signal but assigns zero payments to all its outcomes, eliminating this inspection while appropriately adjusting the remaining payments preserves incentive compatibility and weakly improves the principal's utility. These insights lead to an efficient algorithm.

**Family 2: Settings with independent evaluations.** For settings as in Example 1.2, we establish a key property of optimal contracts—they exhibit a simple structure of payment for at most one signal and one outcome. This structure enables a polynomial-time algorithm that enumerates over single-signal inspection policies. We begin with definitions:

**Definition 3.3** (ISOP). An adaptive contract setting satisfies the *Independent Signal-Outcome Property (ISOP)* if the initial signal $k \sim \mathbf{q}_i^0$ and the inspected outcome $j \sim \mathbf{q}_i^k$ are independent random variables for any action $i \in [n]$. Equivalently, ISOP holds if $\mathbf{q}^k = \mathbf{q}^{k'}$ for all $k, k' \in [\ell]$.

An ISOP setting can be described by a tuple $(\mathbf{q}^0, \mathbf{q}, \mathbf{c}, \mathbf{d}, \mathbf{r})$. The second opinion $j$ is drawn independently from the initial signal $k$, but its distribution matrix $\mathbf{q}$ can be distinct from that of the initial signal $\mathbf{q}^0$, and $j$ has a distinct role from $k$ in determining the reward $r_{k,j}$. However, in some applications the first and second opinions are completely symmetric:

**Definition 3.4** (Symmetric-ISOP). An adaptive contract setting satisfies *Symmetric-ISOP* if (1) it satisfies ISOP; (2) $\mathbf{q}^0 = \mathbf{q}$; and (3) the rewards are symmetric, i.e., $r_{k,j} = r_{j,k}$ for all $j, k \in [\ell]$.

A Symmetric-ISOP setting is a tuple $(\mathbf{q}, \mathbf{c}, \mathbf{d}, \mathbf{r})$, where $\mathbf{r}$ is a symmetric $\ell \times \ell$ matrix. Symmetry strengthens our results, as the following positive result holds for ISOP, while one of our hardness results holds even for Symmetric-ISOP.

**Theorem 3.5.** *Consider adaptive contract settings satisfying ISOP, and target the highest-cost action $n$. Then an optimal adaptive contract for $n$ can be computed in polynomial time. Moreover, without loss of generality, it pays for at most one signal and one outcome.*

# 4. Computational Hardness

In this section we show the necessity of the structural assumptions underlying our algorithms: The design problem becomes intractable with an unbounded number of actions and no ISOP. Full proofs are deferred to the Appendix.

**Theorem 4.1.** *It is NP-hard to approximate the optimal principal's adaptive contracts utility to within any constant.*

*Proof sketch.* We reduce from Independent Set, which is hard to approximate to within any constant factor (Arora & Safra, 1998). The principal wishes to incentivize the agent to take a specific "good" action, but there is also one "bad" action for each edge of the graph. No matter which action the agent chooses, a uniformly random vertex is drawn as the signal. Inspecting a vertex reveals whether the agent took any of the bad actions from the incident edges. The cost of the good action is very low, so assuming there is some nonzero chance of the agent being detected, the principal only needs to pay the agent a tiny transfer to incentivize taking the good action. The main source of cost comes from inspecting vertices. Thus, the optimal contract inspects a minimum vertex cover. The payoff and inspection costs are set up so that the principal's utility is roughly proportional to the number of vertices that are *not* inspected, which is the complement of the minimum vertex cover, i.e., a maximum independent set. See Appendix F.1 for the reduction details. □

We also show that the assumption that our settings satisfy MLRP (see Section 2) is necessary.

**Theorem 4.2.** *If MLRP is violated, then even in settings satisfying Symmetric-ISOP and uniform inspection costs, it is NP-hard to compute the optimal principal's utility.*

*Proof sketch.* We reduce from a variant of Set Cover through an intermediate problem that we call *Row-Sparsest Matrix (RSM)*. The objective of RSM is to fill in an $n \times n$ matrix with nonnegative entries in a way that satisfies a series of linear constraints, while having as few rows with nonzero entries as possible. These matrix entries ultimately become the payments in the optimal contract. The types of linear constraints in RSM take a very restricted form. Most notably, they are symmetric with respect to transposing the matrix, which is why we are able to relate this problem to Symmetric-ISOP. See Appendix F.2 for the details of the proof, which turns out to be quite involved. □

# 5. Beyond Deterministic Inspection

The power of randomization is a ubiquitous phenomenon in algorithmic game theory. In this section we explore it in the context of adaptive contracts. We first show that permitting probabilistic inspection may preclude the existence of a Stackelberg equilibrium in the principal-agent interaction,

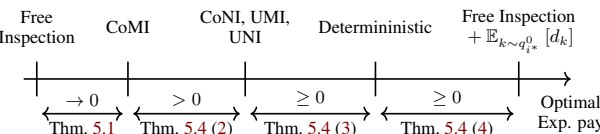

*Figure 2.* Relationships between expected payments in optimal deterministic versus nondeterministic contracts. Note that the strict inequality holds only when the optimal contract inspects.

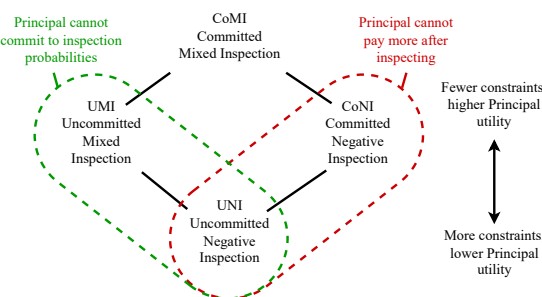

*Figure 3.* Our four nondeterministic problem variants, partially ordered by optimal principal's utility.

since the principal can always deviate to lower-probability inspection with scaled-up payments. Moreover, the resulting contracts which achieve close-to-optimal principal utilities are impractical, as they involve astronomical payments with vanishing probabilities—such contracts are not encountered in practice.[2]

In response to this challenge, we study two practical assumptions that can be imposed separately or together to eliminate this potential deviation by the principal. This results in a total of four problem variants (see Figure 3). In Theorems 5.1 and 5.3, we show that assuming at least one of these is both necessary and sufficient for equilibrium existence. We consider the complexity of the optimal adaptive contract in these variants upper bound the cost savings achieved by randomized inspection. Surprisingly, we show in Theorem 5.6 that there are settings in which randomized inspection can yield higher principal utility even when the principal is unable to commit to inspection probabilities, and thus must be indifferent about whether or not to inspect. In discussing the solutions to the various randomized inspection problem variants, we refer the reader to our QCQP formulation in Appendix D.1, Equation (1).

## 5.1. Committed Mixed Inspection (CoMI)

In the *Committed Mixed Inspection (CoMI)* setting, the principal commits in advance to a random inspection strategy,

---

[2]We are not the first authors to observe this phenomenon. In particular, Mirrlees (1999) derives a similar result in non-adaptive contract setting. Dye (1986) proposes to resolve this issue by assuming risk aversion and bounded payments; we propose two alternative, less-elaborate routes around this paradox.

by assigning each signal a fixed probability of inspection, and then deciding whether to inspect based on a biased coin toss. Formally, optimal CoMI contracts are optimal solutions to Equation (1), under the constraint $p_k \in [0, 1]$ for all $k \in [\ell]$, with $p_k$ interpreted as the probability of inspecting signal $k$ upon observing it. For instance, consider an organization that delegates code generation to an LLM, but remains concerned about potential bugs. To mitigate risk without incurring prohibitive inspection costs, the organization might commit to further inspect at random a fixed percentage (e.g., 10%) of the code modules that pass initial evaluation. Despite the intuitive appeal of this randomized, pre-committed approach as a natural extension of the deterministic model, we show that the CoMI setting does not admit a Stackelberg equilibrium (see Appendix G.2 for the proof). As an immediate corollary, we can resolve the complexity of CoMI.

**Theorem 5.1.** *Given any CoMI instance $X$, let $\widehat{X}$ be the same except with $0$ inspection costs. Then $\widehat{X}$ always has an optimal solution with expected principal utility $y$, and $X$ has an optimal solution if and only if one of the optimal solutions to $\widehat{X}$ incentivizes an action $i$ while never inspecting any signal $k$ such that $q^0_{i,k} d_k > 0$; otherwise, there is a sequence of solutions to $X$ with values converging to $y$.*

**Corollary 5.2.** *There is a polynomial-time algorithm to solve CoMI (in the sense of computing the supremum of possible expected principal utilities).*

*Proof.* Given any CoMI instance $X$, consider the instance $\widehat{X}$ from Theorem 5.1. Since inspection costs zero, it is without loss of generality to set each $p_k = 1$. Then Equation (1) becomes a linear program, so we can solve $\widehat{X}$ in polynomial time to obtain the supremum utility. $\square$

Intuitively, the core argument in the proof of Theorem 5.1 is that for any contract which inspects signals with some probability, the principal can lower their expected payment by decreasing inspection probabilities and proportionally increasing payment for inspected outcomes. This shows that no optimal contract exists under these conditions, and provides a series of contracts that converge towards the optimal value $y$. We note that the resulting contracts approaching the optimal value $y$ become impractical: Their worst-case payment tends to infinity while the probability of receiving any payment tends to zero, exceeding the budget limits of any practical principal, and deterring agents with the slightest degree of risk-aversion. In light of these issues, the next section shows that equilibrium guarantees can be recovered by imposing different restrictions on the principal.

## 5.2. Restoring Equilibrium Guarantees

In this section, we show that equilibrium guarantees can be restored by imposing additional restrictions on the principal.

Specifically, we consider two variants: (1) disallowing commitment to inspection probabilities, and (2) requiring that the payment for inspected outcomes always remains at most that of corresponding uninspected signals. In Theorem 5.3, we prove that each of these restrictions—individually or in combination—restores equilibrium guarantees.

**Uncommitted Mixed Inspection (UMI).** One possible way to restore equilibrium guarantees is to consider settings which generalize beyond the traditional Stackelberg framework. Specifically, we introduce the Uncommitted Mixed Inspection (UMI) variant, in which the principal only commits to payments $(s, t)$, but does not commit in advance to an inspection policy $p$. Unlike the Stackelberg approach taken by classical contract design theory and by the settings we discussed thus far, the objective in UMI is to design a contract $(p, s, t)$ which induces a *subgame-perfect Bayes-Nash equilibrium* of the principal-agent interaction that is optimal for the principal. By inducing such an equilibirium, the principal can declare an intention to inspect a particular signal $k$ with an intermediate probability $0 < p_k < 1$, which the agent will accept as believable. Subgame perfection necessitates that this is credible, meaning the principal must be genuinely indifferent between inspecting and not inspecting the signal. In Appendix G.1, we show how this constraint can be used to simplify Equation (1) into a quadratically-constrained linear program (QCLP), and we discuss how we may practically solve this problem for small instances.

**Committed Negative Inspection (CoNI).** Another possible way to guarantee the existence of an equilibrium is to impose that the payment for an inspected outcome never exceeds the payment for the corresponding uninspected signal. This condition aligns well with scenarios where inspections may uncover negative information, justifying reduced pay. For example, in the code generation use-case discussed in the introduction, further inspection may uncover security vulnerabilities that reduce the code's value. Formally, this is represented by constraints requiring that $t_{k,j} \leq s_k$ for all $k \in [\ell]$ and $j \in [m_k]$, ensuring that inspection can only reduce or maintain the monetary transfer.

**Uncommitted Negative Inspection (UNI).** Finally, we consider a problem variant which integrates both the no-commitment and negative inspection assumptions. Here, the principal does not commit to an inspection policy, and is also subject to the constraint $t_{k,j} \leq s_k$ for all $k$ and $j$.

**Equilibrium Guarantees.** Not every QCQP or QCLP has an optimal value. Hence, it is not clear that any of the three variant games actually have an equilibrium, as there may not be one optimal contract for the principal to choose. The following result, proved in Appendix G.2, shows that optimal contracts exist in all variants:

**Theorem 5.3.** *Any UMI, CoNI, or UNI instance has an optimal solution.*

## 5.3. Deterministic vs. Optimal Inspection

We now turn to the question of how powerful randomized inspection policies can be, in terms of what actions they can incentivize and at what cost. The following theorem characterizes implementability and bounds the cost savings of nondeterminism for CoMI, CoNI, and UMI:

**Theorem 5.4.** *For any adaptive contract setting and action:*

1. *In each of CoMI, CoNI, UMI, and UNI, it is possible to incentivize an action $i$ with a nondeterministic contract if and only if it is possible to incentivize $i$ with a deterministic contract.*

2. *In CoNI, UMI, or UNI, if an optimal contract to incentivize $i$ ever pays for inspection, the principal can incentivize $i$ with a strictly smaller expected payment in CoMI; otherwise, the minimum expected payments are the same.*

3. *In CoNI, UMI, or UNI, the principal's minimal payment to incentivize $i$ is weakly smaller than their minimum payment required under a deterministic contract.*

4. *The infimum expected payment required to incentivize $i$ in CoMI is within $\mathbb{E}_{k \sim q_{i*}^0}[d_k]$ of the minimum payment required to incentivize $i$ in a deterministic contract.*

The relationships between the principal's minimal payment in each variant are depicted in Figure 2, and the proof is in Appendix G.3. This result has complexity implications as well. One might hope that non-deterministic contracts are easier to find since the domain of each $p_k$ variable is the convex set $[0, 1]$ rather than the discrete set $\{0, 1\}$. However, combining Theorem 5.4 (3) and an observation about the proof of Theorem 4.1, we can conclude that at least two of our nondeterministic variants are still hard to approximate:

**Theorem 5.5.** *It is NP-hard to approximate the optimal principal utility to a constant factor in either UMI or UNI.*

See Appendix G.4 for the proof. Finally, we remark that the middle inequality from Figure 2 is *sometimes* strict. In other words, randomized inspection can improve principal's optimal utility in some cases, even with either the commitment or negativity constraints imposed. The proof uses the help of optimization software, and is presented in Appendix G.5.

**Theorem 5.6.** *There exist instances of UMI and CoNI where nondeterministic inspection is strictly optimal.*

## 6. Empirical Evaluation

To demonstrate the value of adaptive contracts in practical scenarios, we present two empirical case studies of adaptive contracts for delegated text generation using LLM evaluation benchmark data. The first case study demonstrates the empirical gains of contracts in a general question answering dataset where our theoretical assumptions do not hold. The second demonstrates the value of adaptivity in a setting

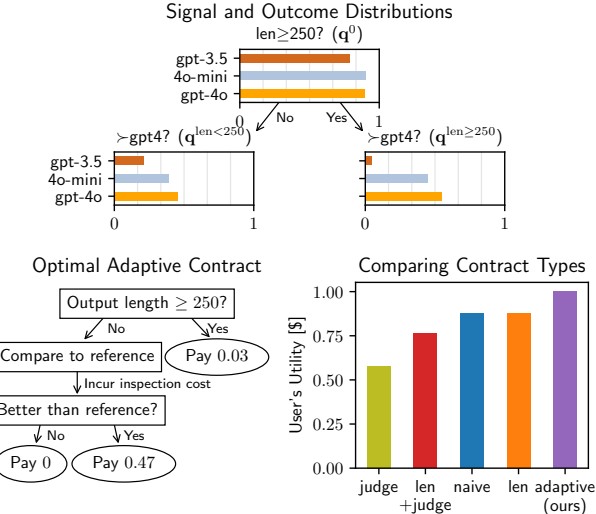

*Figure 4.* Empirical evaluation of adaptive contracts on AlpacaEval data (Section 6.1). **(Top)** Outcome distributions. Initial signal is length thresholding, and inspection provides human-annotated pairwise comparison to a reference output generated by GPT-4. **(Bottom Left)** Optimal adaptive contract. The contract inspects short outputs, and rewards concise answers which are better than the GPT-4 reference. **(Bottom Right)** Comparison of principal utility across different types of contracts. Adaptive contracts yield significantly higher utility to the principal in this setting.

where the assumptions of Theorem 3.5 initially hold, and shows that the functional form of optimal contracts remains simple even as assumptions are gradually relaxed.

### 6.1. Benefits of Adaptivity in Question-Answering

**Dataset.** For our first case study, we use evaluation data gathered from the AlpacaEval 2.0 dataset (Li et al., 2023). AlpacaEval evaluates language models by prompting them with a fixed set of 805 prompts, and comparing responses to outputs generated by a reference model (GPT-4 Turbo). A response is labeled as 'high-quality' if it compares favorably to the reference output, and comparison methods vary.

**Actions, signals and outcomes.** Our setup follows the two-tiered evaluation setting illustrated in Example 1.1: For the space of actions, we use a selection of LLMs currently offered by OpenAI (GPT-3.5 Turbo, GPT-4o Mini, and GPT-4o). As the coarse evaluation signal, we use a response length threshold, which indicates whether the response is longer than 250 characters. As the costly evaluator, we use the AlpacaEval pairwise evaluation, which marks a response as "good-quality" if it is preferred over a reference LLM output generated independently. The induced signal and conditional outcome distributions are illustrated in Figure 4 (Top). Note that the cost of invoking the second evaluation is considerable, as it involves generating a reference output and performing costly pairwise comparison.

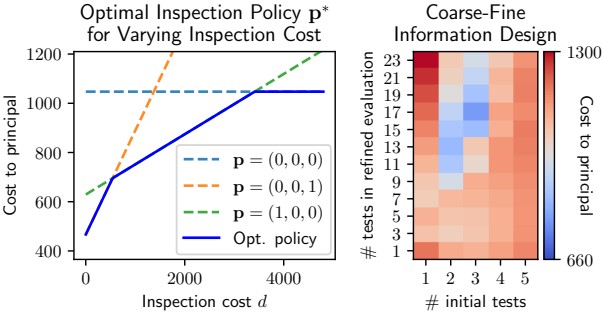

*Figure 5.* Adaptive contracts on SWE-Bench data (Section 6.2). **(Left)** Optimal inspection policy as a function of inspection cost, showing transition between three optimal policies as $d$ grows. **(Right)** Information design optimization heat map. Cells represent cost of optimal contract given initial and refined test counts.

**Costs and rewards.** As a first-order estimate of costs, we use the public OpenAI API pricing, with GPT-4o Mini being the least costly model, and GPT-4o having the highest cost. We also note that the signal distribution $\mathbf{q}^0$ doesn't satisfy MLRP, as GPT-4o Mini tends to generate longer responses compared to GPT-4o, despite being less costly. For inspection, we assume human pairwise comparison costs, and use the human evaluation costs reported by AlpacaEval. For principal rewards, we assume that an answer which compares favorably to the reference GPT-4 output generates a reward of $2, modeling a competitive advantage of over the market baseline. See Section H.2.2 for for further discussion of parameter choice and sensitivity.

**Results.** Results are illustrated in Figure 4. We compute optimal contracts by enumeration, and compare the principal's utility to non-adaptive contract baselines. The optimal contract is illustrated in Figure 4 (Bottom Left): Short responses are inspected, and the contract pays $0.47 for outputs judged as favorable; otherwise, the contract pays $0.03 for long outputs. The adaptive contract incentivizes use of GPT-4o, and yields expected utility of $1.00 to the principal. This utility is %14 higher than the closest non-adaptive baseline (Figure 4 Bottom Right). Appendix H provides further interpretation and details.

### 6.2. Adaptive Contracts in Agentic Coding

**Dataset.** Our second case study analyzes adaptive contracts for delegation of generative coding tasks, based on data from the SWE-Bench benchmark (Jimenez et al., 2024), which evaluates coding agents' ability in complex codebases. We use the 'bash-only' sub-benchmark, which consists of 500 Github issues (bug reports), together with pass/fail unit tests. The goal of the generative model in this benchmark is to resolve the issues by modifying the code to pass the tests.

**Actions, signals and outcomes.** For the space of actions, we use a set of six contemporary OpenAI LLMs, ranging from GPT-3o to GPT-5. In the base setup, the initial signal space represents the success rate in a small suite of 2 tests, and the refined test suite is of size 8. Signal and outcome distributions are approximated by binomial distributions, initialized with the aggregate empirical success rates. As Binomial distributions satisfy MLRP and test results are independent, this setting satisfies both MLRP and ISOP.

**Costs and rewards.** We use the inference costs reported in the SWE-Bench dataset. For inspection costs, we assume that running a single test entails a fixed cost to the principal, and we vary this cost in our experiments. For rewards, we assume that the expected reward of the best-performing model (GPT-5) is high enough so it is always targeted.

**Results and interpretation.** Results are illustrated in Figure 5. As inspection cost $d$ grows, Figure 5 (Left) shows that the optimal contract transitions between three distinct optimal policies: from inspection on full success, to inspection on full failure, and eventually to no inspection. Additionally, Figure 5 (Right) shows a heat-map representing different information-design trade-offs between initial and refined test counts, finding that a suite of 3 initial tests and an extended suite of 17 tests is optimal for the given data.

**Additional experiments.** In Appendix H, we provide additional results and interpretation, showing that simple inspection policies remain optimal even under parameterized correlation or random perturbation of distributions, and analyzing agent utility as informativeness varies.

## 7. Discussion

Our findings open several avenues for future work: First, the problem of finding the optimal adaptive contract is hard with general dimensions and correlated evaluations; are there more nuanced assumptions that make near-optimal adaptive contracts tractable? A natural such assumption is limited correlation among the evaluations. In particular, would it be sufficient for tractability that the outcome is conditionally independent of the signal given a hidden *state-of-nature*?

Second, there is much more to investigate for randomized inspections: We have shown that randomization can strictly lower costs for the user, and that this holds even under either of our two practical constraints (on the user's power to commit to probabilities, and on the magnitude of payments); however, we do not have a tight upper-bound on just how large the cost savings can be. Complexity questions remain as well—we establish either hardness or tractability for three of the four problem variants, but it is unknown whether the final CoNI variant can be solved in polynomial time. Hardness of approximation and hardness subject to assumptions like independence also remain open.

Finally, our setup focuses on a single adaptive round of inspection—a second opinion. Would our positive results for the independent evaluation setting carry over to more than two opinions, e.g., with a cost for each additional opinion or subject to a total inspection budget? Addressing this could shed new light on the general interaction between information acquisition and contract design.

## Impact Statement

This paper presents work whose goal is to advance the field of machine learning by optimizing the allocation of evaluation resources in delegation settings. Our adaptive contracts framework increases the economic efficiency of AI task delegation, potentially reducing the computational and human costs associated with quality assurance. Beyond these contributions to the efficiency and reliability of AI service markets, we are not aware of any societal consequences that require specific discussion.

## Acknowledgments

The authors would like to thank anonymous reviewers for their insightful remarks and valuable suggestions. Eden Saig is supported by the Israel Council for Higher Education PBC scholarship for Ph.D. students in data science, and the Zuckerman STEM Leadership Program. The work of Tamar Garbuz, Eden Saig and Inbal Talgam-Cohen received funding from the European Research Council (ERC) under the European Union's Horizon 2020 research and innovation program (grant No.: 101077862, project: ALGOCONTRACT, PI: Inbal Talgam-Cohen), from the Israel Science Foundation (grant No.: 3331/24), from the NSF-BSF (grant No.: 2021680), and from a Google Research Scholar Award and a Microsoft AIEI Fellowship. Ariel Procaccia was partially supported by the National Science Foundation under grants IIS-2147187 and IIS-2229881; by the Office of Naval Research under grants N00014-24-1-2704 and N00014-25-1-2153; and by a grant from the Cooperative AI Foundation. Jamie Tucker-Foltz was supported by the National Science Foundation Graduate Research Fellowship Program under Grant No. DGE1745303 and a Google PhD Fellowship.

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

## A. Delegated Text Generation as an Adaptive Contract Design Problem

In this section, we formally describe Examples 1.1 and 1.2 as instances of the adaptive contracts framework.

**Delegated text generation.** In an LLM moral hazard setting (see (Saig et al., 2024)), there is a vocabulary of tokens denoted by $V$, and the set of all token sequences is denoted by $V^*$. A *text generator* $g : V^* \to V^*$ is a mapping from a prompt $w_0$ to a response $w_R$. We assume there is a distribution over prompts such that $w_0 \sim \mathcal{D}$, and denote by $\mathcal{D}_g$ the distribution over prompt-response pairs induced by $g$. A *quality evaluator* $q$ is a possibly-stochastic mapping from a prompt-response pair to an integer score. For adaptive evaluation, we assume that there are two quality evaluators $q_L, q_H$, where $q_L$ is coarse and can be evaluated for free, and $q_H$ is more accurate but also costly to invoke.

The agent has access to a set of generators $\mathcal{G} = \{g_1, \dots, g_n\}$, which we also refer to by their indices $[n]$. The expected cost incurred by the agent when using generator $g_i$ is assumed to be proportional to the average response length $c_i = \alpha_i \mathbb{E}_{(\omega_0, \omega_R) \sim \mathcal{D}_i} [\|\omega_R\|]$, where $\alpha_i > 0$ is a generator-dependent coefficient. In contract design terminology, the generators $\mathcal{G}$ are the agent's possible actions. Given a prompt $w_0$, and upon receiving a response $w_R$, the principal invokes the coarse evaluator $q_L$ at no cost, revealing an *initial signal*, denoted by $k = q_L(w_0, w_R) \in [\ell]$. As the coarse signal may be uninformative, the principal may decide to further inspect the response using the costly evaluator $q_H$. Invoking $q_H$ incurs a cost $d_k \geq 0$ for the principal, and reveals an *outcome*, denoted by $j = q_H(w_0, w_R) \in [m]$.

**Two-tiered evaluation.** In Example 1.1, the user first receives a coarse evaluation at no cost, then they may pay for additional refinement. For example, the coarse evaluation may be implemented by length comparison, formally $q_L(w_0, w_R) = 1$ if and only if the length of the response $w_R$ is higher than some fixed threshold. For the costly evaluation, one can use a pairwise comparison method such as LLM-as-a-judge or human evaluators (see Section 6), formally $q_H(w_0, w_R) = 1$ if and only if $w_R$ is preferred to a reference output $w_R^{\text{ref}}$ generated by GPT-4 using the same prompt (Li et al., 2023). Inspection costs $\mathbf{d}$ are induced by the costs of generating text and performing pairwise comparison given the output length. Ex-ante, the signal generated by $g_i \in \mathcal{G}$ is denoted by $k \sim \mathbf{q}_i^0$, and its distribution is induced by the composition of $\mathcal{D}$ and $q_L$. Similarly, the refined outcome is denoted by $j \sim \mathbf{q}_i^k$, and its distribution is induced by composing $\mathcal{D}|k$ with $q_H$.

**Adaptive independent sampling.** In Example 1.2, there is a large batch of issues to be resolved, and the contract rewards the agent based on their performance on a random subset of tasks. Formally, given a batch size $B_0$, the user first samples a batch of tasks, represent as prompts $\{w_0^1, \dots, w_0^{B_0}\} \sim \mathcal{D}^{B_0}$, then a response is generated for each prompt, resulting in a set of prompt-response pairs $W_0 = \{(w_0^1, w_R^1), \dots, (w_0^B, w_R^B)\}$. Initial evaluation is performed by evaluating all input-output pairs in $W_0$ using $q_L$ to yield the initial signal $k$, which represents the joint results, or their aggregate. The signal distribution $\mathbf{q}^0$ is induced by the composition of $\mathcal{D}$, the random sampling that yields $W_0$, and the evaluation $q_L$. Outcome inspection is performed by sampling another (possibly larger) batch $W_1$ of size $B_1$, and evaluating through $q_H$ to yield output $j$. The dependence between $k$ and $j$ vanishes when $B$ is large, and the ISOP assumption increasingly holds (see Definition 3.3).

## B. Limitations

As shown above, the adaptive contract setting significantly generalizes the classic contract setting in a way that enables more efficient delegated text generation contracts. Yet our setting is not without limitations:

**MLRP assumption.** The MLRP assumption is very natural in many settings, as higher evaluation scores are more likely given more costly effort on the provider side. However, not all evaluation methods satisfy this property; for example, the coarse inspection in our experiments (Section 6) simply measures length, and a long response may actually be a sign of a lower-quality, wordy model. We note that the definition of the signal and outcome spaces is to some extent up to the designer—e.g., instead of length thresholding, one could use bucketing, or alternatively, the distance from what would be considered an ideal response length. Such design choices can yield settings that are quite similar, with the additional property of satisfying MLRP.

**Two-level inspection.** Our settings incorporate two levels of inspection, one free and one costly. The reason for this is that this is arguably the simplest possible setting that allows for adaptivity, and it still captures realistic scenarios. But there are scenarios in which a third or even fourth opinion is likely to be solicited; and/or where the inspection policy is best described by a multi-level decision tree. We leave this generalization for future work.

**Deterministic inspection.**    Our setting accommodates both deterministic and randomized inspections, but our positive computational results are for deterministic inspection schemes. We emphasize that this does not mean that the quality evaluators themselves cannot be stochastic; but the decision whether or not to inspect (using the possibly-stochastic evaluator) is binary. The advantage of determinism is that it yields natural adaptive contracts, as demonstrated in Figure 4, where the contract turns out to be "inspect concise answers". Section 5 explores at length the relaxation of deterministic inspection.

**Manipulation of evaluation metrics.**    Our framework assumes that the agent's action space is fixed. However, when the principal employs naive evaluation criteria, the agent may be able to manipulate the outcome distribution without meaningfully altering the true quality of the underlying model. For example, if a contract relies heavily on output length, the agent can tune the model to produce longer answers by tuning model weights, effectively creating a new action "gaming" the contract to maximize expected reward. That said, we also note that many evaluation methods of significant practical interest are in fact robust to gaming. For example, in code generation (discussed in Example 1.1 and Section 6.2), it is generally very challenging to pass unit tests without generating the correct program. As additional examples, direct evaluation by humans (Li et al., 2023) and LLM evaluation based on forecasting (Yang et al., 2025) are both generally robust to this type of manipulation. Our framework supports the full spectrum of evaluation methods, regardless of their susceptibility to manipulation. We hope that our work will further motivate the consideration of strategic elements in LLM evaluation.

**Size of action space.**    Our framework assumes that the space of actions is finite, and the algorithms we presents have polynomial dependence on its cardinality. While the possible number of potential model configurations and hyperparameter is theoretically unbounded, we note that many providers in the AI supply chain operate within a constrained set of alternatives. For example, within the enterprise sector, recent reports indicate that AI services are becoming increasingly embedded within existing business software (Gartner, 2025). Providers of these software products act as intermediate AI providers, processing requests and routing them to a foundation model providers (e.g. OpenAI, Anthropic or Google). The set of actions available to these providers is the set of compatible API endpoints, which is typically finite and reasonable in size: For example, OpenAI currently offers a selection of roughly 40 text models through its API[3], and the OpenRouter platform offers a selection of roughly 400 models through its cross-market API[4]. As a possible direction for future inquiry, we note that the robust contract design results of Carroll (2015) show that it is possible to design non-adaptive incentive-compatible contract even in settings where the principal only has information about a finite and possibly small subset of an action space that may be infinite. We hope that our work will motivate the application of robust approaches to the adaptive setting.

**Bargaining power.**    Our framework assumes that users have the power to negotiate a favorable contract. While this assumption may not fully capture consumer-facing markets with posted-price subscription models, we note that a substantial portion of the AI services market is dominated by business and enterprise clients which are likely to have higher market leverage (Reuters, 2025a;b). Moreover, these enterprise clients typically engage in long-term agreements and have higher vendor lock-in due to the technical integration efforts required. This long-term dependency heightens the risk of strategic quality degradation by AI providers, thereby motivating the consideration of pay-for-performance schemes.

**Common prior.**    The optimization program we present assumes that the score distributions are known by the principal, and remain fixed for the duration of the contract. In response to similar concerns about the strength of these assumptions, the non-adaptive contract design literature offers methods for learning optimal contracts in dynamic environments that may evolve over time (Ho et al., 2014; Zhu et al., 2023), and for robust contract design, which often only requires partial and easier-to-track knowledge about the prior distribution, such as knowledge of expected values instead of full distributions (Dütting et al., 2019). Both of these approaches typically use classic contracts as a basic building block or benchmark, and we hope that our results will motivate their extension to adaptive contract design settings.

## C. Agent's Utility

Our contract design framework primarily adopts a principal-centric perspective, aiming to compute adaptive contracts that maximize the principal's expected utility. However, in many segments of the market, AI pricing and service delivery strategies are currently primarily decided by the model providers, who act as the agents in our setting. To fully contextualize our results within current LLM monetization paradigms, we discuss in this section the changes in the agent's expected utility under adaptive contracts compared to non-adaptive pricing schemes. We find that the introduction of adaptivity can be either

---

[3] https://developers.openai.com/api/docs/pricing
[4] https://openrouter.ai/models

beneficial or detrimental to the agent, depending on the problem parameters, and demonstrate this by identifying two distinct scenarios leading to opposite effects on the agent's utility:

**Adaptive inspection leading to an agent utility decrease.** When inspection is affordable and reveals the model chosen by the agent, then the principal completely eliminates moral hazard by inspecting. This can occur, for example, when outcome distributions of different models have negligible overlap in support, and inspection cost $d$ is sufficiently small. In such cases, the transfer of the optimal contract converges towards the target action's cost, and thus the agent's utility decreases to 0 – effectively achieving the "first-best" solution, with the added cost of inspection. Emprically, we observe this in our code-generation experiments (Section 6.2). As demonstrated in Figure 10 and further detailed in Appendix H.3.2, the agent's utility monotonically decreases as the size of the refined test suite grows, providing the principal with a stronger signal and significantly reducing the moral hazard information gap.

**Adaptive inspection leading to agent utility increase.** Conversely, adaptive contracts can also lead to an increase in the agent's utility. When a high-cost LLM is not cost-effective for the principal without inspection but becomes reward-maximizing with inspection, the agent's utility may increase due to higher information rent. We observe this empirically in our AlpacaEval experiments (Section 6.1), where the most capable and expensive model (GPT-4o) only emerges as the reward-maximizing action for the principal when inspection is permitted. This scenario may be of practical interest, as it demonstrates that adaptive contracts can strictly benefit *both* the users of AI tools (the principal) and AI model providers (the agent), potentially providing additional motivation for adoption of adaptive contracts in practice.

# D. Proof Preliminaries

## D.1. Optimization Program

We seek the action $i$ that maximizes expected reward $R_i$ minus the expected inspection cost $D_i$ and expected payments $T_i$ required to incentivize it. This amounts to solving the following quadratically-constrained quadratic program (QCQP) for each action $i \in [n]$:

$$
\begin{aligned}
\textbf{minimize} \quad & \sum_{k \in [\ell]} q_{i,k}^0 \left( (1 - p_k)s_k + p_k \left( d_k + \sum_{j \in [m_k]} q_{i,j}^k t_{k,j} \right) \right) \\
\textbf{subject to} \quad & s_k \geq 0 && \forall k \in [\ell] \\
& t_{k,j} \geq 0 && \forall k \in [\ell],\ j \in [m_k] \\
& 0 \leq p_k \leq 1 && \forall k \in [\ell] \\
& \sum_{k \in [\ell]} q_{i,k}^0 \left( (1 - p_k)s_k + p_k \sum_{j \in [m_k]} q_{i,j}^k t_{k,j} \right) - c_i \\
& \geq \sum_{k \in [\ell]} q_{i',k}^0 \left( (1 - p_k)s_k + p_k \sum_{j \in [m_k]} q_{i',j}^k t_{k,j} \right) - c_{i'} && \forall i' \in [n]
\end{aligned}
\tag{1}
$$

We refer to the last constraint above as the *IC constraint*.

Note that in the deterministic inspection setting, which is the primary focus of our paper, we additionally require each $p_k$ to take values in $\{0, 1\}$. However, it will be useful to consider the more general problem without this restriction. Even when allowing $p_k$ to take any value in the interval $[0, 1]$, the program remains generally non-convex.

If we fix the values of $p$ and then optimize the remaining $s$ and $t$ variables, the problem reduces to a linear program, as elaborated in the next subsection. This will allow us to treat Equation 1 as a nested optimization problem, and provide tractability guarantees.

## D.2. Combined Outcome Space and Distribution

A key ingredient in our tractability proofs is a correspondence between second opinion contracts and classical contracts for any fixed inspection policy $p$. Given a second opinion contract setting, we define the combined outcome space and

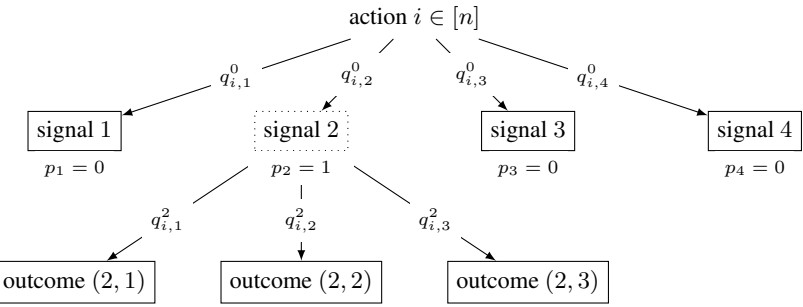

*Figure 6.* Schematic diagram of the combined outcome space $\Omega$. When the inspection policy $p$ is fixed, sampling from the combined outcome distribution $f_{i,\omega}$ (Equation (2)) is equivalent to a random walk on the inspection tree according to the signal and outcome probabilities ($q_{i,j}^0$ and $q_{i,j}^k$, respectively), starting from the root and proceeding until a leaf is reached.

corresponding distributions, which provide a more unified perspective on the signals and outcomes. For $k \in \{0, \dots, \ell\}$, we denote the set of signals by $\Omega_0$ (such that $|\Omega_0| = \ell$), and the sets of outcomes by $\Omega_k$ for $k \geq 1$ (such that $|\Omega_k| = m_k$). We assume that elements in all sets are uniquely labeled, and therefore the sets do not intersect. The *combined outcome space* $\Omega$ is the union:

$$\Omega = \Omega_0 \cup \cdots \cup \Omega_\ell.$$

Given inspection probabilities $p \in [0,1]^\ell$, for any action-outcome pair $(i, \omega)$ the combined outcome distribution is:

$$f_{i,\omega}(p) = \begin{cases} q_{i,\omega}^0 (1 - p_\omega) & \omega \in \Omega_0 \\ q_{i,k}^0 p_k q_{i,\omega}^k & \omega \in \Omega_k \text{ for } k > 0 \end{cases} \tag{2}$$

For brevity, we denote $f_{i,\omega} = f_{i,\omega}(p)$ when $p$ is clear from context. We note that for any action $i$, the vector $f_{i,\omega}$ is a probability distribution since plugging in the definition and rearranging we get:

$$\begin{aligned} \sum_{\omega \in \Omega} f_{i,\omega} &= \sum_{\omega \in \Omega_0} f_{i,\omega} + \sum_{k \in [\ell]} \sum_{\omega \in \Omega_k} f_{i,\omega} \\ &= \sum_{k \in [\ell]} (1 - p_k) q_{i,k}^0 + \sum_{k \in [\ell]} p_k q_{i,k}^0 \underbrace{\sum_{\omega \in \Omega_k} q_{i,\omega}^k}_{=1} \\ &= \sum_{k \in [\ell]} q_{i,k}^0 = 1 \end{aligned}$$

Similarly, the combined payments vector is:

$$v_\omega = \begin{cases} s_\omega & \omega \in \Omega_0 \\ t_{k,\omega} & \omega \in \Omega_k \text{ for } k > 0 \end{cases} \tag{3}$$

Figure 6 provides a schematic representation of the combined outcome space. Using these notations, we reformulate the contract design problem:

**Lemma D.1.** *Equation 1 is equivalent to the following optimization problem:*

$$\begin{aligned} &\textit{minimize} \quad \sum_{\omega \in \Omega} f_{i,\omega} v_\omega + D_i \\ &\textit{subject to} \quad v \geq 0 \\ &\qquad\qquad p \in [0,1]^\ell \\ &\qquad\qquad \sum_{\omega \in \Omega} f_{i,\omega} v_\omega - c_i \geq \sum_{\omega \in \Omega} f_{i',\omega} v_\omega - c_{i'} \quad \forall i' \neq i \end{aligned} \tag{4}$$

*Proof.* By definition of $f_{i,\omega}$ (Equation (2)), $D_i$ (Section 2), and $v_\omega$ (Equation (3)).

The objective of Equation 1 is:

$$\sum_{k\in[\ell]} q_{i,k}^0 \left( (1-p_k)s_k + p_k \left( d_k + \sum_{j\in[m_k]} q_{i,j}^k t_{k,j} \right) \right)$$

Rearranging the sums, we obtain:

$$\underbrace{\sum_{k\in[\ell]} q_{i,k}^0(1-p_k)s_k}_{=\sum_{\omega\in\Omega_0} f_{i,\omega} v_\omega} + \underbrace{\sum_{k\in[\ell]}\sum_{j\in[m_k]} q_{i,k}^0 p_k q_{i,j}^k t_{k,j}}_{=\sum_{k\in[\ell]}\sum_{\omega\in\Omega_k} f_{i,\omega} v_\omega} + \underbrace{\sum_{k\in[\ell]} q_{i,k}^0 p_k d_k}_{=D_i}$$

Which is equal to the objective $\sum_{\omega\in\Omega} f_{i,\omega} v_\omega + D_i$.

Similarly, the (IC) constraint in Equation 1 is:

$$\sum_{k\in[\ell]} q_{i,k}^0 \left( (1-p_k)s_k + p_k \sum_{j\in[m_k]} q_{i,j}^k t_{k,j} \right) - c_i$$

$$\geq \sum_{k\in[\ell]} q_{i',k}^0 \left( (1-p_k)s_k + p_k \sum_{j\in[m_k]} q_{i',j}^k t_{k,j} \right) - c_{i'}$$

Using the same arguments, we rearrange and obtain equivalence to the inequality as required:

$$\sum_{\omega\in\Omega} f_{i,\omega} v_\omega - c_i \geq \sum_{\omega\in\Omega} f_{i',\omega} v_\omega - c_{i'}.$$

$\square$

We make the following observation about Equation 4:

**Lemma D.2.** *For any fixed p, the optimal $v$ in Equation 4 is a MIN-PAY optimal contract.*

*Proof.* In the objective of Equation 4, $\sum_\omega f_{i,\omega} v_\omega$ is the expected pay of the contract $v$, and $D_i$ is an additive term which does not depend on $v$. When $p$ in fixed, $f_i$ and $D_i$ are also fixed. In that case, the constant term $D_i$ does not affect the optimization, and the optimization problem is therefore equivalent to the classic Stackelberg MIN-PAY problem. $\square$

## E. Tractability
### E.1. General Properties of Optimal Contracts

**Proposition E.1.** *Let $(\mathbf{p}, \mathbf{s}, \mathbf{t})$ be a contract. If there exists an inspected signal $k_0 \in [\ell]$ such that $t_{k_0,j} = 0$ for all $j \in [m_{k_0}]$, then there exists a contract $(\mathbf{p}', \mathbf{s}', \mathbf{t}')$ with $p'_{k_0} = 0$ and $p'_k = p_k$ for all $k \neq k_0$, which yields weakly higher utility for the principal.*

*Proof.* Define a modified contract $(\mathbf{p}', \mathbf{s}', \mathbf{t}')$ such that:

$$p'_k = \begin{cases} 0 & \text{if } k = k_0 \\ p_k & \text{otherwise} \end{cases}$$

$$s'_k = \begin{cases} (1-p_{k_0})s_{k_0} & \text{if } k = k_0 \\ s_k & \text{otherwise} \end{cases}$$

$$t'_{k,j} = t_{k,j}$$

For any action $i \in [n]$, the expected monetary transfer from the principal to the agent satisfies:

$$
\begin{aligned}
T_i(\mathbf{p}, \mathbf{s}, \mathbf{t}) &= \sum_{k \in [\ell]} q_{i,k}^0 \left( (1 - p_k)s_k + p_k \sum_{j \in [m_k]} q_{i,j}^k t_{k,j} \right) \\
&= \sum_{k \in [\ell] \setminus \{k_0\}} q_{i,k}^0 \left( (1 - p_k)s_k + p_k \sum_{j \in [m_k]} q_{i,j}^k t_{k,j} \right) \\
&\quad + q_{i,k_0}^0 \left( (1 - p_{k_0})s_{k_0} + p_{k_0} \sum_{j \in [m_{k_0}]} q_{i,j}^{k_0} \underbrace{t_{k_0,j}}_{=0} \right) \\
&= \sum_{k \in [\ell] \setminus \{k_0\}} q_{i,k}^0 \left( (1 - p_k)s_k + p_k \sum_{j \in [m_k]} q_{i,j}^k t_{k,j} \right) \\
&\quad + q_{i,k_0}^0 \underbrace{(1 - p_{k_0})s_{k_0}}_{=s'_{k_0}} \\
&= \sum_{k \in [\ell]} q_{i,k}^0 \left( (1 - p'_k)s'_k + p'_k \sum_{j \in [m_k]} q_{i,j}^k t'_{k,j} \right) \\
&= T_i(\mathbf{p}', \mathbf{s}', \mathbf{t}')
\end{aligned}
$$

The two contracts transfer identical amounts in expectation for any action, and therefore the modified contract $(\mathbf{p}', \mathbf{s}', \mathbf{t}')$ implements the same action. The modified contract $(\mathbf{p}', \mathbf{s}', \mathbf{t}')$ also yields weakly higher utility for the principal, as it does not inspect signal $k_0$. $\square$

### E.2. Constant-Many Actions

In this section we prove Theorem 3.2.

**Lemma E.2.** *Consider a contract design setting $(\mathbf{q}, \mathbf{c}, \mathbf{d})$ with $\ell$ signals and $n$ actions. There is an optimal contract which implements action $i^*$ while inspecting at most $n - 1$ signals (with at most $n - 1$ nonzero payments).*

*Proof.* By contradiction. Denote by $\mathbf{p}^*$ the inspection policy of an optimal contract, and denote the corresponding inspection set by $S = \{k \mid p_k^* > 0\}$. Assume that $\mathbf{p}^*$ has a minimal inspection set size $|S|$ among all optimal contracts, and assume by contradiction that $|S| \geq n$.

By Lemma D.2, for any fixed inspection policy $\mathbf{p} \in [0, 1]^\ell$, and in particular for the given $\mathbf{p}^*$, the optimal signal and outcome payments $\mathbf{s}^*, \mathbf{t}^*$ are equivalent to a MIN-PAY contract over the combined outcome space $\Omega = \Omega_0 \cup \cdots \cup \Omega_\ell$. Denote by $\mathbf{v}^*$ the representation of $\mathbf{s}^*, \mathbf{t}^*$ in the combined outcome space, as given by Equation 3. By (Dütting et al., 2019), a MIN-PAY contract design problem over $n$ actions has an optimal solution with at most $n - 1$ nonzero payments, and therefore we assume that $\mathbf{v}^*$ has at most $n - 1$ nonzero entries. By definition of the combined space $\Omega$, there exist at most $n - 1$ signals $k \in [\ell]$ for which there exists $\omega \in \Omega_k$ such that $v_\omega^* > 0$. Equivalently, in the $\mathbf{s}^*, \mathbf{t}^*$ representation, there exist at most $n - 1$ signals $k \in [\ell]$ for which $t_{k,j}^* > 0$ for some $j \in [m_k]$.

Since $|S| \geq n$, there exists at least one inspected signal $k'$ for which $t_{k',j}^* = 0$ for all $j \in [m_{k'}]$. By Proposition E.1, there exists a weakly-better contract $(\mathbf{p}', \mathbf{s}', \mathbf{t}')$ with a smaller inspection set, contradicting the minimality of $|S|$. Therefore, there exist an optimal contract inspecting at most $n - 1$ signals. $\square$

*Proof of Theorem 3.2.* Denote the set of actions by $[n]$, and consider the following algorithm: For each subset of signals $S \subseteq [\ell]$ such that $|S| \leq n - 1$, compute the optimal payments $\mathbf{s}, \mathbf{t}$ to implement action $i^*$ under the constraint $p_k = \mathbb{1}_{k \in S}$. Output the contract $(\mathbf{p}^*, \mathbf{s}^*, \mathbf{t}^*)$ yielding the minimal expected pay.

By Lemma D.2, for any fixed inspection policy $\mathbf{p} \in [0, 1]^\ell$, the optimization problem for the signal and outcome payments $\mathbf{s}, \mathbf{t}$ is equivalent to a MIN-PAY contract design problem, and therefore the optimal contract in each iteration can be computed

in polynomial time by solving a linear program. There are $O(\ell^n)$ subsets of signals of size smaller than $n$, and since $n = O(1)$, it holds that $O(\ell^n) = O(\text{poly}(\ell))$. Multiplying the time complexity of each iteration by the total number of iterations, we obtain that the algorithm described above runs in $O(\text{poly}(\ell, m))$ time in total.

By Claim E.2, there exists an optimal contract inspecting at most $n - 1$ signals. Since the algorithm enumerates all subsets of signals up to this size, it is guaranteed to encounter the optimal subset of signals, and thus return an optimal result. $\quad\square$

### E.3. ISOP

In this section we prove Theorem 3.5.

**Proposition E.3.** *Consider an adaptive contract setting satisfying ISOP, and targeting the highest-cost action $n$. If the action is implementable, then any optimal contract pays for at most one signal, and for at most one outcome.*

*Proof.* Denote an optimal contract by $(\mathbf{p}^*, \mathbf{s}^*, \mathbf{t}^*)$. Since the setting satisfies ISOP, we denote $q_{i,k}^k = q_{i,k}^1$. We consider two cases:

If the contract doesn't inspect any signal (i.e., $p_k^* = 0$ for all $k \in [\ell]$), then by Lemma D.2, the optimal pay for signals $s^*$ is a MIN-PAY contract over the signal distribution $q_{i,k}^0$ and costs $c_i$. The contract design problem $(q_{i,k}^0, c_i)$ satisfies MLRP, and therefore by (Dütting et al., 2019, Lemma 7) there exists an optimal contract which only pays for the highest signal $k = \ell$.

Otherwise, the contract inspects at least one signal. By Lemma D.2, the transfers $\mathbf{s}^*, \mathbf{t}^*$ are an optimal solution to a MIN-PAY contract design problem over the combined outcome space:

$$
\begin{aligned}
\textbf{minimize} \quad & \sum_{\omega \in \Omega} f_{i,\omega} v_\omega \\
\textbf{subject to} \quad & v \geq 0 \\
& \sum_{\omega \in \Omega} f_{n,\omega} v_\omega - c_n \geq \sum_{\omega \in \Omega} f_{i,\omega} v_\omega - c_i \quad \forall i < n,
\end{aligned}
$$

where $f_{i,\omega}$ is given by Equation (2), and $v_\omega$ is given by Equation (3). By (Dütting et al., 2019), the dual of the min-pay LP is:

$$
\begin{aligned}
\textbf{maximize} \quad & \sum_{i<n} \lambda_i(c_n - c_i) \\
\textbf{subject to} \quad & \lambda \geq 0 \quad\quad\quad\quad\quad\quad\quad\quad\quad\quad\quad\quad\quad\quad\quad (5) \\
& \sum_{i<n} \lambda_i (f_{n,\omega} - f_{i,\omega}) \leq f_{n,\omega} \quad \forall \omega \in \Omega
\end{aligned}
$$

Plugging in the definitions of $f_{i,\omega}$ and $\Omega$ we get:

$$
\begin{aligned}
\textbf{maximize} \quad & \sum_{i<n} \lambda_i(c_n - c_i) \\
\textbf{subject to} \quad & \lambda \geq 0 \\
& \sum_{i<n} \lambda_i \left( q_{n,k}^0 - q_{i,k}^0 \right) \leq q_{n,k}^0 \quad\quad\quad\quad\quad \forall k \text{ s.t. } p_k^* = 0 \\
& \sum_{i<n} \lambda_i \left( q_{n,k}^0 q_{n,j}^1 - q_{i,k}^0 q_{i,j}^1 \right) \leq q_{n,k}^0 q_{n,j}^1 \quad \forall k \text{ s.t. } p_k^* = 1, \ j \in [\ell]
\end{aligned}
$$

Constraints for which $q_{n,k}^0 = 0$ or $q_{n,k}^0 q_{n,j}^1 = 0$ represents signals and outcomes that cannot be reached when the agent takes action $n$. Such constraints are satisfied for any $\lambda$, and are therefore redundant. Ignoring redundant constraints, we divide the first set of constraints by $q_{n,k}^0$ and the second set of constraints by $q_{n,k}^0 q_{n,j}^1$ to obtain:

$$\textbf{maximize} \quad \sum_{i<n} \lambda_i (c_n - c_i)$$

$$\textbf{subject to} \quad \lambda \geq 0$$

$$\sum_{i<n} \lambda_i \left( 1 - \frac{q_{i,k}^0}{q_{n,k}^0} \right) \leq 1 \qquad \forall k \text{ s.t. } p_k^* = 0$$

$$\sum_{i<n} \lambda_i \left( 1 - \frac{q_{i,k}^0}{q_{n,k}^0} \cdot \frac{q_{i,j}^1}{q_{n,j}^1} \right) \leq 1 \quad \forall k \text{ s.t. } p_k^* = 1, \, j \in [\ell]$$

From the MLRP assumption, the ratio $\frac{q_{i,k}^0}{q_{n,k}^0}$ is decreasing in $k$. We denote $k_0^* = \max\{k \mid p_k^* = 0\}$. For any $i < n$, $k < k_0^*$, and $\lambda_i \geq 0$, it holds that:

$$\lambda_i \left( 1 - \frac{q_{i,k}^0}{q_{n,k}^0} \right) \leq \lambda_i \left( 1 - \frac{q_{i,k_0^*}^0}{q_{n,k_0^*}^0} \right)$$

and therefore the first set of constraints satisfies:

$$\sum_{i<n} \lambda_i \left( 1 - \frac{q_{i,k_0^*}^0}{q_{n,k_0^*}^0} \right) \leq 1 \quad \Rightarrow \quad \sum_{i<n} \lambda_i \left( 1 - \frac{q_{i,k}^0}{q_{n,k}^0} \right) \leq 1 \tag{6}$$

Similarly, we denote $k_1^* = \max\{k \mid p_k^* = 1\}$. For any $i < n$, $k < k_1^*, j \in [\ell]$, and $\lambda_i \geq 0$, it holds that:

$$\lambda_i \left( 1 - \frac{q_{i,k}^0}{q_{n,k}^0} \cdot \frac{q_{i,j}^1}{q_{n,j}^1} \right) \leq \lambda_i \left( 1 - \frac{q_{i,k_1^*}^0}{q_{n,k_1^*}^0} \cdot \frac{q_{i,\ell}^1}{q_{n,\ell}^1} \right)$$

and therefore the second set of constraints satisfies:

$$\sum_{i<n} \lambda_i \left( 1 - \frac{q_{i,k_1^*}^0}{q_{n,k_1^*}^0} \cdot \frac{q_{i,\ell}^1}{q_{n,\ell}^1} \right) \leq 1 \quad \Rightarrow \quad \sum_{i<n} \lambda_i \left( 1 - \frac{q_{i,k}^0}{q_{n,k}^0} \cdot \frac{q_{i,j}^1}{q_{n,j}^1} \right) \leq 1 \tag{7}$$

From Equation (6) and Equation (7), the dual LP has at most two binding constraints. Equation (5) is thus equivalent to:

$$\textbf{maximize} \quad \sum_{i<n} \lambda_i (c_n - c_i)$$

$$\textbf{subject to} \quad \lambda \geq 0$$

$$\sum_{i<n} \lambda_i \left( 1 - \frac{q_{i,k_0^*}^0}{q_{n,k_0^*}^0} \right) \leq 1$$

$$\sum_{i<n} \lambda_i \left( 1 - \frac{q_{i,k_1^*}^0}{q_{n,k_1^*}^0} \cdot \frac{q_{i,\ell}^1}{q_{n,\ell}^1} \right) \leq 1$$

And therefore from complementary slackness, the optimal solution for the primal LP pays for at most one signal, and at most one outcome. $\qquad \square$

*Remark* E.4. Unlike the classical result of (Dütting et al., 2019) where the optimal contract pays only for the highest outcome, in our case, the optimal contract does not necessarily follow this structure (restrict payment to the highest outcome of the highest signal). Instead, it pays for at most the highest not inspected signal and at most the highest outcome of the highest inspected signal. This distinction separates our model from the classical setting.

*Proof of Theorem 3.5.* By Proposition E.3, any optimal contract $(\mathbf{p}^*, \mathbf{s}^*, \mathbf{t}^*)$ pays for one signal at most, and one outcome at most. Therefore, by Proposition E.1 it can be assumed without loss of generality that in the optimal contract there exists at most one $k \in [\ell]$ such that $p_k = 1$. All single-signal inspection policies can be enumerated in linear time, and an optimal contract for each policy can be computed in polynomial time, yielding a polynomial-time algorithm for the whole computation. $\square$

## F. Hardness

### F.1. Proof of Theorem 4.1 (Hardness of Approximation Without ISOP)

We reduce from Independent Set. Consider an input graph $G = (V, E)$, and arbitrarily pick a small constant $\varepsilon \in (0, 1)$. We assume that vertices and edges are indexed and uniquely labeled, and denote by $\mathrm{adj}(e) = \{u, v\}$ the nodes adjacent to a given edge $e \in E$. We define a contract design instance as follows:

- **Actions, signals, and outcomes.** The set of actions is $E \cup \{\mathrm{target}\}$, and the set of signals is $V \cup \{\mathrm{dummy}\}$, where $\{\mathrm{target}, \mathrm{dummy}\}$ are special symbols. Thus, the number of actions is $n = |E| + 1$, and the number of signals is $\ell = |V| + 1$. Each signal is associated with a binary inspected outcome space, such that $m_k = 2$ for all signals $k \in [\ell]$. To simplify presentation, we denote here $[m_k] = [2] = \{A, B\}$.

- **Action and inspection costs.** Actions corresponding to edges $e \in E$ cost $c_e = 0$. The cost of the target action is $c_{\mathrm{target}} = \varepsilon \left(|V| + 1\right)^{-1}$. The cost of inspection for any vertex signal $v \in V$ is $d_v = |V| + 1$. The cost of inspection for the dummy signal is $d_{\mathrm{dummy}} = \left(|V| + 1\right)^2$.

- **Signal and outcome distributions.** For all actions, the signal distributions $\mathbf{q}_i^0$ are uniform, formally $q_{i,k}^0 = \left(|V| + 1\right)^{-1}$ for all action-signal pairs $(i, k) \in [n] \times [\ell]$. When vertex signals $v \in V$ are inspected under edge actions $e \in E$, the outcome distribution is deterministic, and given by:

$$\mathbf{q}_e^v = \begin{cases} (1, 0) & v \in \mathrm{adj}(e) \\ (0, 1) & \text{otherwise} \end{cases}$$

When vertex signals $v \in V$ are inspected under the target action, the outcome distribution is:

$$\mathbf{q}_{\mathrm{target}}^v = (0, 1)$$

For the dummy signal, the inspected outcome distribution is $(0, 1)$ for the target action, and $(1, 0)$ otherwise. Note that this clearly satisfies Inspection-MLRP, as the distribution over signals is the same regardless of the chosen action, while for any signal, the more costly action makes the $B$ outcome weakly more likely.

- **Principal rewards.** The reward for the signal-outcome pair $(\mathrm{dummy}, B)$ is $\left(|V| + \varepsilon\right)\left(|V| + 1\right)$. The reward for any other signal-outcome pair is zero.

**Lemma F.1.** *The expected reward of the target action is $R_{\mathrm{target}} = |V| + \varepsilon$, and the expected reward for any other action $e \in E$ is $R_e = 0$.*

*Proof.* By definition, the only signal-outcome pair which yields reward is $(\mathrm{dummy}, B)$. When the target action is taken, the dummy signal has probability $\left(|V| + 1\right)^{-1}$, yielding a total expected reward of $R_{\mathrm{target}} = |V| + \varepsilon$. For any other action, the probability of obtaining this signal-outcome pair is zero, and therefore the total reward for any other action is zero as well. $\square$

**Lemma F.2.** *Let $G = (V, E)$ be a graph, and let $\varepsilon \in (0, 1)$. For the corresponding contract design instance. Denote the set of inspected signals by $S$, and by $f(S)$ the optimal expected pay of the principal when signals $S$ are inspected (expected transfer and inspection costs). Then it holds that $f(S) \in [|S|, |S| + \varepsilon)$ if $S$ is a vertex cover of $G$, and $f(S) \geq |V| + 1$ otherwise.*

*Proof.* Consider the contract that pays $s_k = 0$ for all signals, and $t_v = (0, \varepsilon)$ for all inspected signals $v \in S$. For any edge

action $e \in E$, the agent's utility satisfies:

$$U_A(\text{target}) - U_A(e) = \sum_{v \in S} q^0_{\text{target},v} t_{v,B} - \underbrace{c_{\text{target}}}_{=\frac{\varepsilon}{|V|+1}} - \sum_{v \in S \setminus \text{adj}(e)} q^0_{e,v} t_{v,B} + \underbrace{c_e}_{=0}$$

$$= \frac{\varepsilon}{|V|+1} \underbrace{(|S| - |S \setminus \text{adj}(e)|)}_{\geq 1 \text{ as } S \text{ covers } e} - \frac{\varepsilon}{|V|+1}$$

$$\geq 0$$

and therefore the contract implements the target action. The expected cost of inspection is:

$$D_{\text{target}} = \sum_k q^0_{\text{target},k} p_k d_k$$

$$= \frac{|S|}{|V|+1} (|V|+1)$$

$$= |S|$$

The expected monetary transfer from the principal to the agent is non-negative by definition, and it also holds that:

$$T_{\text{target}} \leq \sum_k q^0_{\text{target},k} \left( (1 - p_k) s_k + p_k \sum_j q^k_{\text{target},j} t_{k,j} \right)$$

$$= \frac{|S|}{|V|+1} \varepsilon$$

$$< \varepsilon$$

and hence $f(S) = T_{\text{target}} + D_{\text{target}} = [|S|, |S| + \varepsilon)$ as required.

Conversely, if the set of inspected signals is not a vertex cover of $G$, we first note that the dummy signal must be inspected, because otherwise there exist some edge $e$ which is not covered by the set of inspected signals, and the combined outcome distribution of action $e$ is identical to the combined outcome distribution of the target action, and is the action $e$ therefore an infeasibility witness. Then, if the set of inspected signals contains the dummy signal (i.e., $p_{\text{dummy}} = 1$), then the expected cost of inspection satisfies $\mathbb{E}[p_k d_k] \geq q^0_{i,\text{dummy}} d_{\text{dummy}} = |V| + 1$ as required. $\square$

Returning to the proof of Theorem 4.1, it follows from (Arora & Safra, 1998) that it is NP-hard, for any constant $\alpha > 1$, to distinguish between the following two cases for any given graph $G$ and integer $k$:

**YES instance**: $G$ contains an independent set of size at least $k$.

**NO instance**: Every independent set of $G$ has size strictly less than $k/\alpha$.

Suppose toward a contradiction that we can approximate the optimal principal utility to within a factor of $\alpha$ in polynomial time. Then we could distinguish between the YES and NO instances as follows:

- In the YES case, there exists an independent set $I \subseteq V$ with $|I| \geq k$. Hence, the complement set $V \setminus I$ is a vertex cover with size $|V| - |I| \leq |V| - k$. By Lemma F.2, choosing to inspect signals corresponding to this vertex cover yields a principal's optimal expected payment satisfying

$$f(V \setminus I) \leq |V| - k + \varepsilon.$$

Thus, by Lemma F.1, the optimal principal's utility (reward minus pay) is at least

$$R_{\text{target}} - f(V \setminus I) \geq (|V| + \varepsilon) - (|V| - k + \varepsilon) = k.$$

Since our approximation algorithm achieves an $\alpha$-approximation, it must return a solution with expected principal utility at least $k/\alpha$.

- In the NO case, every independent set has size less than $k/\alpha$, hence the smallest vertex cover has size at least $|V| - k/\alpha + 1$. By Lemma F.2, any set that is not a vertex cover incurs an expected pay of at least $|V| + 1$, making it clearly non-optimal. Therefore, the optimal solution corresponds to inspecting a vertex cover of size at least $|V| - k/\alpha + 1$, and thus the principal's optimal utility is at most

$$R_{\text{target}} - (|V| - k/\alpha + 1) = (|V| + \varepsilon) - (|V| - k/\alpha + 1) = k/\alpha - 1 + \varepsilon < k/\alpha.$$

Any solution returned by the algorithm in this case must therefore have utility strictly less than $k/\alpha$.

Thus, we can distinguish the YES and NO cases, contradicting the hardness of approximating Independent Set. □

## F.2. Proof of Theorem 4.2 (Hardness of Symmetric-ISOP without MLRP)

We show hardness via a series of two reductions from a variant of Set Cover. The intermediate problem is a promise problem which we call *Row-Sparsest Matrix*, or *RSM* for short. An instance of RSM is a tuple $(G, A, y)$, where $G = (V, E)$ is a simple, undirected graph on $|V| = n$ vertices numbered $1, 2, \ldots, n$ with no isolated vertices, with $A \subseteq V$ and $y \in [n]$. The objective is to distinguish the following two cases:

**YES instance.** There exists an $n \times n$ matrix $M$ of nonnegative real numbers such that:

1. The average value in $M$ is 1.

2. At most $y$ rows of $M$ contain nonzero values.

3. For all $\{i, j\} \in E$, we have $M_{i,i} = M_{i,j} = M_{j,i} = M_{j,j} = 0$.

4. For all $k \in A$, $\sum_{i \in [n]} M_{i,k} + \sum_{i \in [n]} M_{k,i} \geq 10n$

**NO instance.** There does *not* exist an $n \times n$ matrix $M$ of nonnegative real numbers such that:

1. The average value in $M$ is between 1 and $\frac{12}{11}$.

2. At most $y$ rows of $M$ contain values greater than or equal to $\frac{5}{11}$.

3. For all $\{i, j\} \in E$, we have $M_{i,i}, M_{i,j}, M_{j,i}, M_{j,j} \leq \frac{5}{11}$.

4. For all $k \in A$, $\sum_{i \in [n]} M_{i,k} + \sum_{i \in [n]} M_{k,i} \geq n$.

**Lemma F.3.** *The problem RSM is NP-hard.*

*Proof.* We reduce from the variant of Set Cover where the number of elements is restricted to be exactly $\frac{1}{9}$ the number of sets. (Set Cover remains NP-hard with this restriction since we can duplicate elements and/or sets until the equality is satisfied.) The input to Set Cover consists of a collection of elements $x_1, x_2, \ldots, x_m$, a collection of sets of elements which we number $S_{m+1}, S_{m+2}, \ldots, S_n$ and a target value $y \in [n - m]$. The objective is to determine whether a collection of at most $y$ of the sets covers all $m$ elements. By our assumption on the number of elements versus the number of sets, we have $n = 10m$.

Given such an instance, let $H$ be the bipartite graph where there is an edge between $i \in [n]$ and $j \in [n]$ if $i \leq m, j > m$, and element $x_i$ is contained in set $S_j$. We then output the RSM instance $(G, A, y)$ where $G$ is the complement of $H$ and $A = [m]$. We will show that, if a set cover of size $y$ exists, then $(G, A, y)$ is YES instance; otherwise, it is a NO instance.

First suppose $C$ is a set cover of size $y$. For each element $x_i$, let $M_{j,i} = 10n$ for one arbitrary $j$ such that $x_i \in S_j \in C$ (which must exist since $C$ is a set cover). Let all other entries of $M$ be zero. Observe that $M$ satisfies all four YES instance properties:

1. The sum of all nonzero elements is $10n \cdot m = n^2$, so the average value is 1.

2. Only the $y$ rows corresponding to the sets in the cover have nonzero entries.

3. We have a nonzero value at $(i, j)$ only when $\{i, j\}$ is an edge in $H$. This does not include any edges of $G$ (since $G$ is the complement of $H$) or diagonal entries.

4. For each $i \in [m]$, we know that, for some $j$, $M_{j,i} = 10n$.

Conversely, suppose $(G, A, y)$ is *not* a NO instance. This means there *does* exists a matrix $M$ satisfying all of the NO instance properties. Let $I$ be the set of indices $i \in [m]$ such that the $i^{\text{th}}$ row of $M$ contains values greater than or equal to $\frac{1}{2}$; Let $J$ be the set of such indices from $m + 1$ to $n$. For each $i \in I$, choose an arbitrary column $j$ such that $M_{i,j}$ is at least $\frac{1}{2}$. Let $f : I \to [n]$ be the mapping of arbitrary choices for each such index $i$. Consider the collection of sets

$$C := \{S_{f(i)} \mid i \in I\} \cup \{S_j \mid j \in J\}.$$

Note that $|C| \le |I| + |J| \le y$ from property (2). We claim that $C$ is a set cover. Consider an arbitrary element $x_i$. We know that there must exist some $j \in [n]$ such that $M_{j,i} \ge \frac{1}{2}$ or $M_{i,j} \ge \frac{1}{2}$, for otherwise we would have

$$\sum_{j \in [n]} M_{j,i} + \sum_{j \in [n]} M_{i,j} < 2n \cdot \frac{1}{2} = n,$$

violating property (4). Furthermore, from the definition of $G$, we know that $j$ must be an index greater than $m$, and $S_j$ contains $x_i$, for otherwise both cases $M_{j,i} \ge \frac{1}{2}$ and $M_{i,j} \ge \frac{1}{2}$ would violate property (3). In the former case ($M_{j,i} \ge \frac{1}{2}$), we have $j \in J$, so $x_i \in S_j \in C$. In the latter case ($M_{i,j} \ge \frac{1}{2}$), we have $i \in I$, so $x_i \in S_{f(i)} \in C$. $\qquad\square$

Returning to the proof of Theorem 4.2, we reduce from RSM, which is NP-hard by Lemma F.3. Given an instance $(G, A, y)$ where $G = (V, E)$ has $n$ vertices, we define an instance with $n + 1$ signals/outcomes, numbered $0, 1, 2, \ldots, n$. Let

$$\varepsilon := 1 - \sqrt{\frac{(n-1)^2(n+2)}{n^3}} > 0. \tag{8}$$

The reward for any realization $(k, j)$, where $k, j \in \{0, 1, 2, \ldots, n\}$, is $\frac{1}{\varepsilon}$ times the number of vertices realized (i.e. nonzero values among $k$ and $j$). All signals cost $\frac{1}{11}$ to get a second opinion, which will be second independent sample from the same probability distribution.

There are three kinds of actions the agent can take, enumerated as follows.[5]

- Action $g$ (the *good action*), which costs 1 and leads to a uniformly random vertex.

- For each vertex $k \in A$, an action $a_k$ which costs 1. With probability $\varepsilon$, outcome 0 is realized; and with probability $1 - \varepsilon$, a uniformly random vertex *other than $k$* is realized.

- For each edge of $G$, between vertex $i$ and vertex $j$, an action $e_{i,j}$, which costs 0. With probability $\varepsilon$, outcome 0 is realized; with probability $\frac{1-\varepsilon}{2}$, $i$ is realized; and with probability $\frac{1-\varepsilon}{2}$, vertex $j$ is realized.

We will show that, if $(G, A, y)$ is a YES instance, there exists a contract where the principal can attain expected utility at least

$$U := \frac{2}{\varepsilon} - 1 - \frac{y}{11n};$$

Whereas if it is a NO instance, there does not exist such a contract. We derive an equivalent interpretation of this utility target as follows. If the agent takes any action other than the good action $g$, a vertex is realized each draw with probability at most $1 - \varepsilon$, so the expected reward of the principal is at most

$$2 \cdot (1 - \varepsilon) \cdot \frac{1}{\varepsilon} = \frac{2}{\varepsilon} - 2 < U.$$

Hence, it is not possible for the principal to attain expected utility at least $U$ if the agent is taking any action besides $g$. On the other hand, when the agent takes action $g$, the probability of realizing a vertex is 1 each draw, so the principal's expected reward is

$$2 \cdot \frac{1}{\varepsilon} = U + 1 + \frac{y}{11n}.$$

Thus, the principal can obtain expected utility at least $U$ if and only if it is possible to incentivize the agent to take action $g$ by paying at most $1 + \frac{y}{11n}$ in expectation.

---

[5]To maintain consistency in indices used for vertices throughout the proof, we break with the convention that $i$ is for actions, $j$ is for outcomes, and $k$ is for signals.

For the forward direction, suppose $(G, A, y)$ is a YES instance. Consider the contract $t$ that pays $M_{i,j}$ when vertex $i$ is realized on the first draw and vertex $j$ is realized on the second draw. Any time outcome $0$ is observed, the payment is zero. By property (2), all but $y$ rows of $M$ are all-zero, meaning they do not require inspection: Upon observing a vertex $i$ for which the $i^{\text{th}}$ row of $M$ is all-zero, the principal will simply pay zero and not inspect. Thus the total inspection cost is

$$(\text{num inspected outcomes}) \cdot (\text{probability of outcome}) \cdot (\text{inspection cost}) = y \cdot \frac{1}{n} \cdot \frac{1}{11} = \frac{y}{11n}.$$

Additionally, the total expected transfer from the principal to the agent under the good action $g$ is

$$T_g = \sum_{i \in [n]} \sum_{j \in [n]} \frac{1}{n^2} M_{i,j} = 1.$$

Hence, when the agent takes the good action $g$, the principal's total expected payment is $1 + \frac{y}{11n}$ as desired. It remains to check the incentive constraints, that $g$ is optimal for the agent.

Each of the $e_{i,j}$ actions yields a transfer of $0$, as the only possible outcomes are combinations of $i$, $j$, and $0$, which all result in zero payment by property (3). Thus, $e_{i,j}$ costs one less than $g$ but also earns one less, so the agent weakly prefers $g$.

Next consider an action $a_k$. We may lower-bound the difference between the expected transfer $T_g$ if the agent chooses $g$ and $T_{a_k}$ if the agent chooses $a_k$ as follows:

$$
\begin{aligned}
T_g - T_{a_k} &= \frac{1}{n^2} M_{k,k} + \sum_{i \in [n] \setminus \{k\}} \left( \frac{1}{n^2} \right) (M_{i,k} + M_{k,i}) + \sum_{i,j \in [n] \setminus \{k\}} \left( \frac{1}{n^2} - \frac{(1-\varepsilon)^2}{(n-1)^2} \right) M_{i,j} \\
&= \frac{1}{n^2} M_{k,k} + \left( \frac{1}{n^2} \right) \sum_{i \in [n] \setminus \{k\}} (M_{i,k} + M_{k,i}) + \left( -\frac{2}{n^3} \right) \sum_{i,j \in [n] \setminus \{k\}} M_{i,j} \\
&\qquad \text{(rearranging Equation (8))} \\
&\geq \frac{1}{n^2} M_{k,k} + \frac{1}{5n^2} \sum_{i \in [n] \setminus \{k\}} (M_{i,k} + M_{k,i}) - \frac{2}{n^3} \sum_{i,j \in [n]} M_{i,j} \\
&\qquad \text{(since } \frac{4}{5n^2} \geq \frac{2}{n^3} \text{ for } n \geq 3) \\
&\geq \frac{2}{5n^2} M_{k,k} + \frac{1}{5n^2} \sum_{i \in [n] \setminus \{k\}} (M_{i,k} + M_{k,i}) - \frac{2}{n^3} \sum_{i,j \in [n]} M_{i,j} \\
&= \frac{1}{5n^2} \left( \sum_{i \in [n]} M_{i,k} + \sum_{i \in [n]} M_{k,i} \right) - \frac{2}{n^3} \sum_{i,j \in [n]} M_{i,j} \\
&\geq \frac{1}{5n^2} (10n) - \frac{2}{n^3} \sum_{i,j \in [n]} M_{i,j} \quad \text{(from property (4))} \\
&= \frac{2}{n} \left( 1 - \frac{1}{n^2} \sum_{i,j \in [n]} M_{i,j} \right) = 0,
\end{aligned}
$$

where in the final equality we have used property (1). Thus, the agent earns weakly more from taking action $g$ than action $a_k$. Since the two actions cost the same, the agent weakly prefers $g$.

For the backward direction, suppose there is a contract in which the principal incentivizes action $g$ by paying at most $1 + \frac{y}{11n}$. Our objective is to show that $(G, A, y)$ is *not* a NO instance. Let $M$ be the matrix where $M_{i,j}$ is the payment upon realizing vertex $i$ from the first draw and $j$ from the second draw. Note that $M$ is constant on rows corresponding to vertex indices that are not inspected. We will show that $M$ satisfies all four NO instance properties.

First, observe that, since the contract incentivizes the costly action $g$, it must transfer at least $T_g \geq 1$ in expectation from the principal to the agent, otherwise the agent is better off taking one of the actions that costs $0$. It follows that the principal can only inspect $y$ actions, for otherwise the total cost exceeds $1 + \frac{y}{11n}$. On the other hand,

$$T_g \leq 1 + \frac{y}{11n} \leq 1 + \frac{1}{11} = \frac{12}{11}.$$

This proves property (1), since $T_g$ is precisely the average value of the matrix $M$.

For any edge $\{i,j\}$ and any entry $z \in \{M_{i,i}, M_{i,j}, M_{j,i}, M_{j,j}\}$, we have

$$
\begin{aligned}
z &< \frac{5(1-\varepsilon)^2}{4} z \quad \text{(for large enough $n$)} \\
&\leq \frac{5(1-\varepsilon)^2}{4}(M_{i,i} + M_{i,j} + M_{j,i} + M_{j,j}) \\
&= 5T_{e_{i,j}} \\
&\leq 5(T_g - 1) \quad \text{(since the contract incentivizes $g$)} \\
&\leq 5\left(\frac{12}{11} - 1\right) \quad \text{(from the inequality above)} \\
&= \frac{5}{11}.
\end{aligned}
$$

This establishes property (3). Furthermore, since each row $i$ is constant if $i$ is not inspected, and $M_{i,i} < \frac{5}{11}$ (each diagonal entry appears as some $z$ in the argument above because we assume $G$ has no isolated vertices), we have that the entire row $i$ must be less than $\frac{5}{11}$ if $i$ is not inspected. As only $y$ rows are inspected, property (2) follows.

Finally, for each $k \in A$, from the incentive constraint that the agent prefers action $g$ to $a_k$, we have

$$
\begin{aligned}
0 &\leq T_g - T_{a_k} \\
&\leq \frac{1}{n^2} M_{k,k} + \left(\frac{1}{n^2}\right) \sum_{i \in [n] \setminus \{k\}} (M_{i,k} + M_{k,i}) + \left(-\frac{2}{n^3}\right) \sum_{i,j \in [n] \setminus \{k\}} M_{i,j} \\
&\qquad \text{(the inequality is only due to the fact that the contract may pay for outcome 0)} \\
&\leq \frac{4}{n^2} M_{k,k} + \frac{2}{n^2} \sum_{i \in [n] \setminus \{k\}} (M_{i,k} + M_{k,i}) - \frac{2}{n^3} \sum_{i,j \in [n]} M_{i,j} \quad \left(\text{since } \frac{2}{n^2} \geq \frac{2}{n^3}\right) \\
&= \frac{2}{n^2}\left(\sum_{i \in [n]} M_{i,k} + \sum_{i \in [n]} M_{k,i} - nT_g\right) \\
&\leq \frac{2}{n^2}\left(\sum_{i \in [n]} M_{i,k} + \sum_{i \in [n]} M_{k,i} - n\right).
\end{aligned}
$$

This implies property (4):
$$
\sum_{i \in [n]} (M_{i,k} + M_{k,i}) \geq n. \qquad \square
$$

## G. Beyond Deterministic Inspection

### G.1. Computing Optimal Randomized Contracts in UMI

In view of the QCQP in Equation (1) which can be used to solve CoMI, we may handle UMI by imposing the following additional subgame-perfection constraints:

- For each $k \in [\ell]$ such that $p_k < 1$,

$$
s_k \leq d_k + \sum_{j \in [m_k]} q_{i,j}^k t_{k,j}. \tag{9}
$$

- For each $k \in [\ell]$ such that $p_k > 0$,

$$
s_k \geq d_k + \sum_{j \in [m_k]} q_{i,j}^k t_{k,j}. \tag{10}
$$

Observe that it is without loss of generality to enforce the constraint given by Equation 9, even when $p_k = 1$. This is because the variable $s_k$ is irrelevant to both the objective function and the other constraints, so we may satisfy this constraint by setting $s_k := d_k + \sum_{j \in [m_k]} q_{i,j}^k t_{k,j}$. We would like to similarly expand the constraint given by Equation 10. However, when $p_k = 0$ it is *not* without loss of generality to assume constraint Equation 10 holds, because if we were to apply the same trick it could require setting some $t_{k,j}$ to be negative. To circumvent this issue, suppose we fix a set of signals $S_0$ with the additional rule that $p_k = 0$ for all $k \in S_0$, compatible with a computational approach of enumerating all subsets of signals. Then we may enforce constraint Equation 10 for all $k$, as long as we remove the stipulation that $t_{k,j} \geq 0$ for $k \in S_0$. This lets us combine constraints Equation 9 and Equation 10 into a single equation:

$$s_k = d_k + \sum_{j \in [m_k]} q_{i,j}^k t_{k,j}.$$

This equality must hold for all $k \in [\ell]$, which allows us to simplify both the objective function and the constraints, and remove all occurrences of the $s_k$ variables. Unfortunately, it does not let us remove all of the quadratic terms in the IC constraint. The result is thus a quadratically-constrained linear program (QCLP), parameterized by an action $i$ and a set of signals $S_0$:

$$
\begin{aligned}
\textbf{minimize} \quad & \sum_{k \in [\ell]} q_{i,k}^0 d_k + \sum_{k \in [\ell]} q_{i,k}^0 \sum_{j \in [m_k]} q_{i,j}^k t_{k,j} \\[1em]
\textbf{subject to} \quad & \sum_{j \in [m_k]} q_{i,j}^k t_{k,j} \geq -d_k && \text{for all } k \in S_0 \\
& t_{k,j} \geq 0 && \text{for all } k \in \overline{S_0}, j \in [m_k] \\
& p_k = 0 && \text{for all } k \in S_0 \\
& 0 \leq p_k \leq 1 && \text{for all } k \in \overline{S_0} \\
& \sum_{k \in [\ell]} q_{i,k}^0 \left( (1 - p_k)d_k + \sum_{j \in [m_k]} q_{i,j}^k t_{k,j} \right) - c_i \geq \\
& \quad \sum_{k \in [\ell]} q_{i',k}^0 \Big( (1 - p_k) \Big( d_k + \sum_{j \in [m_k]} q_{i,j}^k t_{k,j} \Big) && \text{for all } i' \in [n] \\
& \qquad + p_k \sum_{j \in [m_k]} q_{i',j}^k t_{k,j} \Big) - c_{i'}
\end{aligned}
$$

By enumerating all sets of signals $S_0$, we may effectively optimize small UMI instances (as we do for the proof of Theorem 5.6).

### G.2. Proofs of Theorems 5.1 and 5.3 (Characterization of Equilibrium Existence)

We will require the following lemma for both proofs.

**Lemma G.1.** *Fix any feasible solution $(\mathbf{p}, \mathbf{s}, \mathbf{t})$ to a QCQP from CoMI and a signal $k \in [\ell]$ such that $p_k \in (0, 1]$. For any $p_k' \in (0, p_k)$, there is some value of $s_k'$ and vector $\mathbf{t}_k' = (t_{k,1}', t_{k,2}', \ldots, t_{k,m_k}')$ giving a feasible instance $(\mathbf{p}', \mathbf{s}', \mathbf{t}')$, where $\mathbf{p}'$, $\mathbf{s}'$, and $\mathbf{t}'$ are the same as $\mathbf{p}$, $\mathbf{s}$, and $\mathbf{t}$ except on indexes involving signal $k$. Furthermore:*

1. *If $q_{i,k}^0 d_k > 0$, then $(\mathbf{p}', \mathbf{s}', \mathbf{t}')$ has a strictly better objective value than $(\mathbf{p}, \mathbf{s}, \mathbf{t})$; specifically, the principal's expected payment changes by $q_{i,k}^0 (p_k' - p_k) d_k < 0$.*

2. *The map $p_k' \mapsto (s_k', t_k')$ is continuous over the open interval $(0, p_k)$.*

*Proof of Lemma G.1.* For each $j \in [m_k]$, we define

$$s_k' := \frac{1 - p_k}{1 - p_k'} \cdot s_k, \qquad\qquad t_{k,j}' := \frac{p_k}{p_k'} \cdot t_{k,j}.$$

This is clearly continuous over the open interval $(0, p_k)$. Observe that, for all $i' \in [n]$ (including $i' = i$, where $i$ is the specified action the principal is trying to incentivize), we have

$$
\begin{aligned}
(1 - p_k')s_k' + p_k' \sum_{j \in [m_k]} q_{i',j}^k t_{k,j}' &= (1 - p_k')\frac{1 - p_k}{1 - p_k'} s_k + p_k' \sum_{j \in [m_k]} q_{i',j}^k \frac{p_k}{p_k'} t_{k,j} \\
&= (1 - p_k)s_k + p_k \sum_{j \in [m_k]} q_{i',j}^k t_{k,j}
\end{aligned}
\tag{11}
$$

Note that this equality holds for all signals (by definition), not just the specific $k$ for which the variables changed. It follows that the IC constraint still holds. All of the other constraints obviously continue to hold as well.

In what follows, we use a signal index variable $k'$ to avoid clashing with the specific signal $k$ from the theorem statement. The objective value of $(\mathbf{p}', \mathbf{s}', \mathbf{t}')$ is

$$\sum_{k' \in [\ell]} q_{i,k'}^0 \left( (1 - p_{k'}') s_{k'}' + p_{k'}' \left( d_{k'} + \sum_{j \in [m_{k'}]} q_{i,j}^{k'} t_{k',j}' \right) \right)$$

$$= \sum_{k' \in [\ell]} q_{i,k'}^0 \left( (1 - p_{k'}') s_{k'}' + p_{k'}' \sum_{j \in [m_{k'}]} q_{i,j}^{k'} t_{k',j}' \right) + \sum_{k' \in [\ell]} q_{i,k'}^0 p_{k'}' d_{k'}$$

$$= \sum_{k' \in [\ell]} q_{i,k'}^0 \left( (1 - p_{k'}) s_{k'} + p_{k'} \sum_{j \in [m_{k'}]} q_{i,j}^{k'} t_{k',j} \right) + \sum_{k' \in [\ell]} q_{i,k'}^0 p_{k'}' d_{k'} \quad \text{(by Equation (11) above)}$$

$$= \sum_{k' \in [\ell]} q_{i,k'}^0 \left( (1 - p_{k'}) s_{k'} + p_{k'} \sum_{j \in [m_{k'}]} q_{i,j}^{k'} t_{k',j} \right) + \sum_{k' \in [\ell]} q_{i,k'}^0 p_{k'} d_{k'} + \sum_{k' \in [\ell]} q_{i,k'}^0 (p_{k'}' - p_{k'}) d_{k'}$$

$$= \sum_{k' \in [\ell]} q_{i,k'}^0 \left( (1 - p_{k'}) s_{k'} + p_{k'} \left( d_{k'} + \sum_{j \in [m_{k'}]} q_{i,j}^{k'} t_{k',j} \right) \right) + \sum_{k' \in [\ell]} q_{i,k'}^0 (p_{k'}' - p_{k'}) d_{k'}.$$

Since the first sum in the final line above is the objective value of $(\mathbf{p}, \mathbf{s}, \mathbf{t})$, we see that the difference is

$$\sum_{k' \in [\ell]} q_{i,k'}^0 (p_{k'}' - p_{k'}) d_{k'}.$$

Each of the terms in this sum is zero by definition, except for $k' = k$. Thus, the difference in objective values is precisely $q_{i,k}^0 (p_k' - p_k) d_k$ as claimed. □

*Proof of Theorem 5.1.* Fix a CoMI instance $X$. First note that the instance $\widehat{X}$ with zero inspection costs has an optimal solution because it is without harm to inspect all signals, so we may set $p_k = 1$ for all signals, obtaining a linear program.

We first prove the second part of the claim, that there is a sequence of solutions to $X$ whose value converges to the optimal value of $\widehat{X}$. Let $(\mathbf{p}, \mathbf{s}, \mathbf{t})$ be an optimal solution with value $y$. Let $S$ be the set of signals $k \in [\ell]$ such that $p_k > 0$, and suppose that the minimum nonzero $p_k$ value is $\delta$. In $X$, which has a different objective function but the same feasible set, the value of $(\mathbf{p}, \mathbf{s}, \mathbf{t})$ is $y + \sum_{k \in S} q_{i,k}^0 p_k d_k$. For any $0 < \varepsilon < \delta$, we repeatedly apply Lemma G.1 to each signal $k \in S$ such that, obtaining a new solution which improves the objective value by an additive $\sum_{k \in S} q_{i,k}^0 (\varepsilon - p_k) d_k$. Thus, the new solution, call it $(\mathbf{p}^\varepsilon, \mathbf{s}, \mathbf{t}^\varepsilon)$, has objective value

$$y + \sum_{k \in S} q_{i,k}^0 p_k d_k + \sum_{k \in S} q_{i,k}^0 (\varepsilon - p_k) d_k = y + \varepsilon \sum_{k \in S} q_{i,k}^0 d_k$$

Sending $\varepsilon \to 0$, we see that the value of $(\mathbf{p}^\varepsilon, \mathbf{s}, \mathbf{t}^\varepsilon)$ converges to $y$, as desired.

We now prove the characterization of when $X$ has an optimal solution. Clearly, if $\widehat{X}$ has an optimal solution with $p_k = 0$ for all $k$ such that $q_{i,k}^0 d_k > 0$, then that same solution yields the same objective value in $\widehat{X}$. This is optimal because the minimum objective value in $X$ is necessarily at least the minimum objective value of $\widehat{X}$. Conversely, suppose that $X$ has an optimal solution $(\mathbf{p}, \mathbf{s}, \mathbf{t})$ of some value $y'$. Then notice that $y' = y$ by the claim proved in the previous paragraph, since there are certainly solutions to $X$ of value arbitrarily close to $y$. This means that $(\mathbf{p}, \mathbf{s}, \mathbf{t})$ is an optimal solution to $\widehat{X}$, and it cannot possibly inspect any signal $k$ for which $q_{i,k}^0 d_k > 0$ with nonzero probability, for otherwise its objective value would be different in $X$ and $\widehat{X}$. □

*Proof of Theorem 5.3.* In all three problem variants, every QCQP/LCQP has a continuous objective function. Furthermore, the feasible set is closed, as all constraints are specified by weak inequalities. Hence, as long as the feasible set is also bounded, an optimal solution exists, as it is the result of minimizing a continuous function over a compact set. Note that all relevant variables are bounded below, and the $p_k$ variables are bounded above (by one). Thus, an optimal solution is guaranteed to exist as long as each $s_k$ and $t_{k,j}$ is bounded above. We will show that, in each of UMI, CoNI, and UNI, we may impose additional constraints upper-bounding these variables without harming the value of the optimal solution.

We begin with UMI and UNI, where we saw that it is possible to rewrite each QCLP so that it only involves variables $t_{k,j}$ and not any $s_k$. Fix an action $i \in [n]$ and a set of non-inspected signals $S_0 \subseteq [\ell]$. For each $k \in [\ell]$ and $j \in [m_k]$, there are two cases to consider. If $q_{i,k}^0 \cdot q_{i,j}^k = 0$, then variable $t_{k,j}$ irrelevant to the game, so we may set it to zero without loss of generality. Otherwise, if $q_{i,k}^0 \cdot q_{i,j}^k > 0$, then we may upper-bound $t_{k,j}$ by $(\max_{i' \in [n]} R_{i'})/(q_{i,k}^0 q_{i,j}^k)$, for if $t_{k,j}$ is greater than this quantity, then the expected payment from the principal to the agent is more than $(\max_{i' \in [n]} R_{i'})$. This means the principal's utility is negative, so the principal would have been better off with all-zero payments. Thus, in either case, we may upper bound all variables without harming the optimal objective value.

We next consider CoNI. By similar reasoning, we first claim that we may upper bound each $t_{k,j}$ by 0 (in the case where $q_{i,k}^0 \cdot q_{i,j}^k = 0$ for the given action $i$) or $(\max_{i' \in [n]} R_{i'})/(q_{i,k}^0 q_{i,j}^k)$ (otherwise). To see why the latter bound still holds, observe that, if it is violated, then the objective function contains the term

$$q_{i,k}^0 \left((1-p_k)s_k + p_k \left(d_k + q_{i,j}^k t_{k,j}\right)\right) \geq q_{i,k}^0 \left((1-p_k)t_{k,j} + p_k q_{i,j}^k t_{k,j}\right) \geq q_{i,k}^0 q_{i,j}^k t_{k,j} > \max_{i' \in [n]} R_{i'},$$

where in the first inequality we have used the negative inspection constraint and dropped the non-negative $d_k$ term, and in the second inequality we have used the fact that $q_{i,j}^k \leq 1$. So as before, the principal would have been better off with all-zero payments.

Having established that the $t_{k,j}$ variables are bounded, we next proceed to bound the $s_k$ variables. Specifically, we claim that it is without harm to the objective value to bound

$$s_k \leq \max_{j \in [m_k]} U_{k,j},$$

where $U_{k,j}$ is our previous upper bound on $t_{k,j}$. Suppose this inequality is violated. If $p_k = 0$, then we may easily again derive that the principal's payment is more than $\max_{i' \in [n]} R_{i'}$. Otherwise, Lemma G.1 implies we can improve our objective value by locally decreasing $p_k$ and increasing several $t_{k,j}$. Since $s_k$ is strictly larger than the largest $t_{k,j}$, a small enough perturbation will still respect the negative inspection constraint. □

### G.3. Proof of Theorem 5.4

We will show that it is possible to transform each of a deterministic, CoNI, UMI, and CoMI contract into one another while incentivizing the same action $i$, which will prove (1). We will also bound the gaps between the minimum expected payments of each transformation, proving the other three statements.

First suppose there is some deterministic contract incentivizing action $i$. We will show that there is an equivalent contract in UNI (and thus in CoNI and UMI as well). Specifically, for each signal $k$, we will transform the $s_k$ and $t_{k,j}$ variables in a way that satisfies the commitment/negativity constraints and does not change the principal's expected payment. For signals $k$ such that $p_k = 0$, the $t_{k,j}$ variables are payoff-irrelevant. Thus, we may set $t_{k,j} := s_k$ to satisfy both the negativity constraint and the relevant commitment constraint, namely Equation (9). Likewise, for all signals $k$ such that $p_k = 1$, the $s_k$ variables are payoff-irrelevant, so we may set $s_k = d_k + \max_j t_{k,j}$ to both the negativity constraint and the relevant commitment constraint, namely Equation (10). Thus, we can transform any deterministic contract into an UNI contract with the same expected payment, so the optimal CoNI/UMI/UNI contracts have weakly lower expected payments. This proves (3).

Next take an arbitrary contract in CoNI or UMI incentivizing action $i$. Since such a contract is also a valid solution to CoMI, the principal's minimum expected payment in CoMI is weakly smaller. If, additionally, the contract sometimes pays for a second opinion, there must be some signal $k$ such that $q_{i,k}^0 > 0$, $p_k > 0$, and $d_k > 0$. By Lemma G.1 (1) (stated and proved in Appendix G.2), we can decrease the inspection probability of signal $k$ and adjust payments accordingly to get a strictly smaller payment in CoMI. This proves (2).

Finally, to complete the cycle, take an arbitrary CoMI contract $(\mathbf{p}, \mathbf{s}, \mathbf{t})$ incentivizing action $i$. Consider the alternative contract $(\mathbf{p}', \mathbf{s}', \mathbf{t}')$ where each $p'_k = 1$, $s'_k$ is defined arbitrarily, and

$$t'_{k,j} = (1 - p_k)s_k + p_k t_{k,j}.$$

In other words, $(\mathbf{p}', \mathbf{s}', \mathbf{t}')$ simulates $(\mathbf{p}, \mathbf{s}, \mathbf{t})$ by always inspecting every signal and sometimes simply ignoring the outcome and using the old $s_k$ payments. The agent's expected utilities for each action are clearly the same in $(\mathbf{p}', \mathbf{s}', \mathbf{t}')$ as in $(\mathbf{p}, \mathbf{s}, \mathbf{t})$, so $(\mathbf{p}', \mathbf{s}', \mathbf{t}')$ still incentivizes action $i$. The additional expected payment is

$$\sum_{k \in [\ell]} q^0_{i,k}(1 - p_k)d_k \leq \sum_{k \in [\ell]} q^0_{i,k} d_k = \mathbb{E}_{k \sim q^0_i}\left[d_k\right],$$

which proves (4). $\qquad\square$

### G.4. Proof of Theorem 5.5

We observe that the reduction from Vertex Cover in Theorem 4.1 gives us the same bounds on the principal's utility in UMI and UNI:

- Suppose that $G$ contains a large independent set. Then we can apply the reduction to obtain a deterministic contract with high utility. By Theorem 5.4 (3), the principal's utility can only be greater in UMI or UNI than in the deterministic contract.

- Suppose there is a UMI/UNI contract yielding high utility for the principal. Define $S$ to be the set of vertices inspected with nonzero probability. The exact same argument as before shows that $S$ is a vertex cover. To prove that $S$ is small, the key observation is that, since the principal is unable to commit to probabilities, the principal must weakly prefer paying the inspection cost for the signal of each vertex in $C$. Hence, the cost to the principal per vertex in $S$ is just as high, so the size of $S$ must be small. We thus obtain the same lower bound on the size of the complement of $S$, which is an independent set. $\qquad\square$

### G.5. Proof of Theorem 5.6

All numerical claims in this proof have been verified computationally, using Gurobi (Gurobi Optimization, LLC, 2024) to solve the various non-convex programs discussed previously. Consider an instance with three actions, two signals, and two outcomes per signal, with

$$\mathbf{q}^0 = \begin{pmatrix} 0.5 & 0.5 \\ 0.6 & 0.4 \\ 0.6 & 0.4 \end{pmatrix}; \quad \mathbf{q}^1 = \mathbf{q}^2 = \begin{pmatrix} 0.6 & 0.4 \\ 0.4 & 0.6 \\ 0.6 & 0.4 \end{pmatrix}$$

$$\mathbf{c} = \begin{pmatrix} 0 & 0 & 1 \end{pmatrix}; \quad \mathbf{d} = \begin{pmatrix} 1 & 1 \end{pmatrix}.$$

Suppose the rewards are such that the principal wishes to incentivize the costly action 3. Since action 3 is positively correlated with signal 1 and outcome 1, the optimal deterministic contract inspects signal 1 and only pays for outcome 1. The necessary payment is $16 + \frac{2}{3}$, and the total expected cost (including inspection) is $6.6$. However, with a nondeterministic contract, it is preferable to save on inspection cost by inspecting randomly. In CoNI, the optimal inspection probability is $0.625$, for an expected cost of $6.375$; in UMI, the optimal probability is $0.525$, for an expected cost of $6.315$. $\qquad\square$

## H. Experiment Details

### H.1. Implementation

**Code.** We implement our analysis in Python. Our experiments use `cvxpy` for solving contract design linear programs (Diamond & Boyd, 2016), and `matplotlib` for plotting (Hunter, 2007).

Code is available at `https://github.com/edensaig/adaptive-contracts`.

**Hardware and Runtime.** Analyses were run on a single Macbook Pro laptop, with 16GB of RAM, M2 processor, and no GPU. A single run of the data analysis pipeline takes roughly ten minutes to complete. Most of the computation time is devoted towards bootstrap repetitions of the sensitivity analysis (see below).

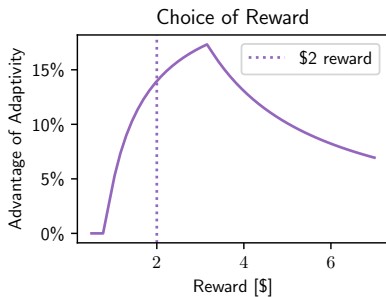
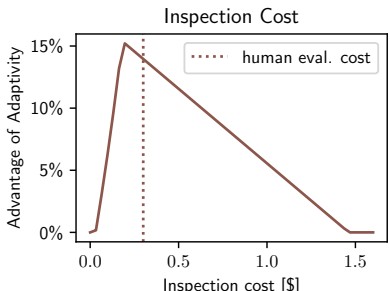
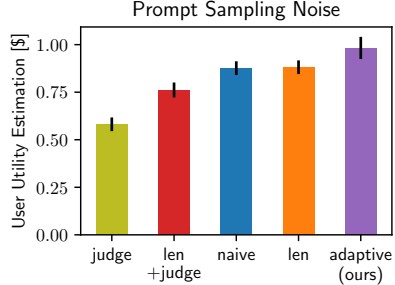

*Figure 7.* Sensitivity analysis for AlpacaEval experiment, as described in Section H. **(Left)** Sensitivity to choice of reward, measured as the relative difference between the optimal adaptive contract and the optimal non-adaptive contract for each parameter choice. **(Center)** Sensitivity to inspection cost. **(Right)** Sensitivity to prompt sampling noise via bootstrapping. Error bars represent the standard deviation.

## H.2. AlpacaEval

### H.2.1. CONTRACT DESIGN PARAMETERS

**Action set.** For the set of actions, we use three recent LLMs provided by OpenAI, designated in the AlpacaEval dataset as: `{gpt-3.5-turbo-1106, gpt-4o-mini-2024-07-18, gpt-4o-2024-05-13}`.

**Signal and outcome distributions.** Signal and outcome distributions are estimated using the empirical proportions in the dataset (length comparison for the coarse signal, and comparison to a GPT-4 reference as the refined evaluation). The corresponding distribution matrices are given below, and additionally illustrated in Figure 4 (Top). Right column of each matrix represents the probability of the positive signal or outcome:

$$\mathbf{q}^0 = \mathbf{q}^{\text{len} \geq 250?} = \begin{pmatrix} 0.21 & 0.79 \\ 0.10 & 0.90 \\ 0.11 & 0.89 \end{pmatrix} ;$$

$$\mathbf{q}^1 = \mathbf{q}^{\succ \text{GPT-4? } | \text{ len} < 250} = \begin{pmatrix} 0.78 & 0.22 \\ 0.61 & 0.39 \\ 0.54 & 0.46 \end{pmatrix} ; \quad \mathbf{q}^2 = \mathbf{q}^{\succ \text{GPT-4? } | \text{ len} \geq 250} = \begin{pmatrix} 0.95 & 0.05 \\ 0.56 & 0.44 \\ 0.45 & 0.55 \end{pmatrix}$$

**Costs and rewards.** We estimate the agent's internal cost of response by multiplying the mean response length by the current OpenAI API prices:[6] \$1.5/1M tokens for GPT-3.5, \$0.6/1M tokens for GPT-4o Mini, and \$10/1M tokens for GPT-4o. We estimate the number of tokens using the "4 characters per token" rule of thumb given in the OpenAI tokenizer guidelines.[7] For the cost of second-tier evaluation, we use the the human evaluation costs reported by AlpacaEval as reference (\$300/1K examples). The \$2 reward was chosen as an order-of-magnitude estimate of online retail profit margin; see sensitivity analysis in Section H.2.2. Overall, the resulting problem parameters are:

$$\mathbf{c} = \begin{pmatrix} 0.00030 & 0.00028 & 0.00468 \end{pmatrix} ; \quad \mathbf{d} = \begin{pmatrix} 0.3 & 0.3 \end{pmatrix} ; \quad R_i = \begin{pmatrix} 0.17 & 0.88 & 1.08 \end{pmatrix}$$

**Baselines.** In Figure 4, we compare performance to the following non-adaptive baselines:

- **Naive.** The `naive` baseline models the interaction between a strategic provider, and a user which doesn't consider moral hazard risks. The contract pays the constant cost of the most expensive model, but receives the reward associated with the least expensive model, as the provider has no incentive to invest additional effort.

- **Only coarse evaluation.** The `len` baseline performs only coarse evalutation by comparing the output length to a fixed threshold at no cost. In the adaptive contracts framework, this is equivalent to a contract that never inspects.

- **Only costly evaluation.** The `judge` baseline represents a setting in which the user only performs the costly evaluation.

- **Always inspect.** The `len+judge` baseline represents a non-adaptive setting in which the user always inspects.

---

[6]OpenAI API pricing: https://platform.openai.com/docs/pricing; archived version in supplementary materials.
[7]OpenAI tokenizer: https://platform.openai.com/tokenizer

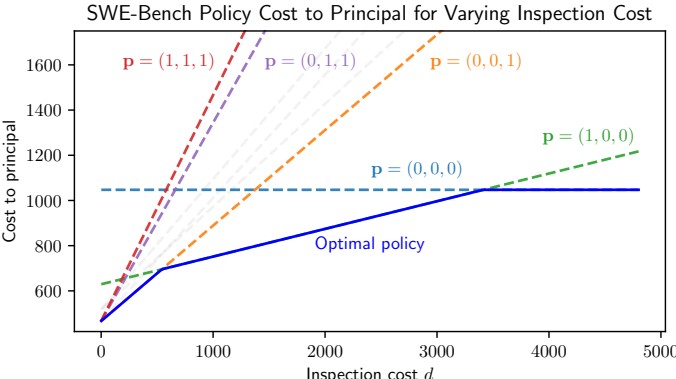

*Figure 8.* Adaptive contracts on SWE-Bench data for different policies $\mathbf{p}$ varying inspection cost $d$, presenting all possible inspection policies. Policies which are optimal for some value of $d$ are colored and labeled, and the optimal inspection policy is designated with a solid line, and policies that suboptimal for any $d$ are represente by dashed gray lines. Compare to Figure 5 (Left), and see Appendix H.3.2.

### H.2.2. SENSITIVITY ANALYSIS

We provide additional numerical experiments to assess the sensitivity of our results to the choice of economic parameters and the distribution of prompts:

**Size of reward.** Figure 7 (Left) presents the expected user utility $U_P$ of adaptive contracts in comparison to the best non-adaptive contract, as a function of the reward parameter. We observe a unimodal curve with strictly positive support for all rewards above some threshold, suggesting that adaptivity is beneficial in many regions of the parameter space, and most beneficial for certain parameter values: When the reward are too low compared to inspection costs, inspection costs are higher than rewards, and therefore no inspection is beneficial, and the optimal contract converges with the non-adaptive contract which never inspects. Conversely, when rewards become very large, the cost of inspection become small in comparison, and the advantage over non-adaptive contracts that always inspects is also vanishing. This suggests the existence of a "sweet spot", and we can indeed see that the advantage of adaptivity in our experiment peaks when the reward is around \$3. We also note that the optimization of adaptive contracts includes all non-adaptive contracts in its optimization space, so the user always weakly benefits from treating the design problem as adaptive.

**Cost of inspection.** Figure 7 (Center) presents the expected advantage of adaptive contracts as a function of the inspection cost $\mathbf{d}$. Similar to the reward sensitivity curve, a unimodal behavior is observed here as well: When inspection costs approach zero, it is always beneficial to inspect all signals, and there is no advantage over the non-adaptive contract which always inspects. Conversely, when the inspection costs are high, there is no advantage over the contract that never inspects. In our experiments, the advantage of adaptivity remains positive even when the inspection costs are increased or decreased by up to five-fold.

**Prompt sampling noise.** Figure 7 (Right) presents an estimation of performance uncertainty due to prompt sampling noise, using the bootstrapping method. At each repetition, we sample prompts with replacement from the dataset, extract problem parameters from the corresponding LLM responses in the dataset, and compute optimal contracts and user utilities for each baseline. The base step was repeated 100 times. Figure 7 (Right) presents mean values and standard deviations, suggesting statistical significance of adaptivity performance gains in our setting with respect to prompt sampling noise. Quantitatively, adaptive contracts exhibit strictly favorable performance in $90 \pm 5.9\%$ of cases at 95% confidence level (binomial proportion confidence interval), and also note that the advantage of adaptive contracts is always non-negative, as non-adaptive contracts are a special case of adaptive contracts. Conditioning on repetitions where performance is strictly favorable, the average relative performance gain is in the interval $[1.5\%, 21\%]$ at %95 confidence level, computed by taking the advantage statistic quantiles over bootstrap repetitions.

## H.3. SWE-Bench

### H.3.1. CONTRACT DESIGN PARAMETERS

For the set of actions, we use six recent LLMs provided by OpenAI:

| SWE-Bench designation | Model name | Success rate $\mu_i$ | Cost $c_i$ |
|---|---|---|---|
| 20250807_MINI-V1.7.0_GPT-OSS-120B | gpt-oss-120b | 0.26 | 28.56 |
| 20250807_MINI-V1.7.0_GPT-5-NANO | GPT-5 nano (2025-08-07) (medium reasoning) | 0.348 | 19.038 |
| 20250726_MINI-V1.0.0_O4-MINI-2025-04-16 | o4-mini (2025-04-16) | 0.45 | 104.99 |
| 20250726_MINI-V1.0.0_O3-2025-04-16 | o3 (2025-04-16) | 0.584 | 166.83 |
| 20250807_MINI-V1.7.0_GPT-5-MINI | GPT-5 mini (2025-08-07) (medium reasoning) | 0.598 | 17.739 |
| 20250807_MINI-V1.7.0_GPT-5 | GPT-5 (2025-08-07) (medium reasoning) | 0.65 | 140.19 |

We extract data from the `metadata.yaml` files in the `bash-only` directory of the SWE-Bench experiments repository[8]. For an adaptive setup with $\ell$ initial signals and $m$ inspected outcomes, we set:

$$\mathbf{q}_i^0 = \text{Binomial}(\ell - 1, \mu_i) \qquad \mathbf{q}_i^1 = \text{Binomial}(m - 1, \mu_i) \tag{12}$$

where $\mu_i$ is the success rate specified above. Note that a $\text{Binomial}(k, p)$ distribution has $k + 1$ possible outcomes $0, \ldots, k$. For inspection costs, denote by $\delta$ the expected cost of running a single test. For an adaptive contracts with $\ell$ signals and $m$ outcomes, the inspection cost is $d_k = \delta(m - 1)$ for all $k \in [\ell]$, and we add $\delta(\ell - 1)$ to the total expected pay to account for the cost of running the initial tests. In Figure 5 (Left) we let $\delta$ vary, and in Figure 5 (Right) we set $\delta = 125$.

### H.3.2. ADDITIONAL EMPIRICAL EVALUATION

In this section, we extend the empirical analysis in Section 6.2. We first provide further interpretation of the role of inspection cost, and then present two additional experiments showing that simple inspection policies, which inspect at most one signal, remain optimal even after introducing parameterized correlation or random perturbations of the distribution matrices. Finally, we empirically study how the agent's utility varies with the degree of information asymmetry.

**Multiple Optimal Policies at Zero Inspection Cost.** Figure 5 (Left) presents the optimal inspection policy $\mathbf{p}^*$ as a function of inspection cost $d$, restricted to optimal policies that inspect at most one signal to simplify graphical presentation. For completeness, we note that multiple policies are optimal when inspection cost is zero ($d = 0$). This is consistent with our theoretical findings: the contract design setting in Section 6.2 satisfies MLRP+ISOP, and therefore by Theorem 3.5 the optimal adaptive contract pays for at most one signal and one outcome. Then, Theorem E.1 guarantees the existence of an optimal adaptive contract which inspects one signal at most. This implies that there exists an optimal inspection policy $p$ with support of size at most 1 for all $d \geq 0$. And indeed, we obtain that $\mathbf{p} = (0, 0, 1)$, $\mathbf{p} = (0, 1, 1)$ and $\mathbf{p} = (1, 1, 1)$ are all optimal for $d = 0$, whereas $\mathbf{p} = (0, 0, 1)$ is the unique optimal inspection policy for any strictly positive and sufficiently small inspection cost.

**Beta-Binomial correlation.** To model interdependence between tests, we utilize a Beta-Binomial compound distribution. Beta-Binomial is an extension of the Binomial distribution, in which the binomial success probability has a prior beta distribution. In the context of adaptive contracts, we use the beta prior to introduce correlation between tests: As initial observations (observed signal $k$) provide information about the value of $\mu$ (and thus about the conditional outcome distributions) via Bayes' rule. Formally, following the formulation of Hisakado et al. (2006) we introduce a correlation coefficient $\rho \in (0, 1)$. Given the original expected value $\mu_i$, we define the modified signal distribution as:

$$\tilde{\mathbf{q}}_i^0 = \text{BetaBinomial}\left(\ell - 1, \left(\frac{1 - \rho}{\rho}\right)\mu_i, \left(\frac{1 - \rho}{\rho}\right)(1 - \mu_i)\right) \tag{13}$$

Which gives an expected success rate of $\mu_i$, consistent with Equation (12). Since the beta distribution is a conjugate prior, and the modified distribution of each $\tilde{\mathbf{q}}_i^k$ is given by the Beta-Binomial posterior:

$$\tilde{\mathbf{q}}_i^k = \text{BetaBinomial}\left(m - 1, k + \left(\frac{1 - \rho}{\rho}\right)\mu_i, \ell - 1 - k\left(\frac{1 - \rho}{\rho}\right)(1 - \mu_i)\right) \tag{14}$$

---

[8]https://github.com/SWE-bench/experiments/tree/main/evaluation/bash-only

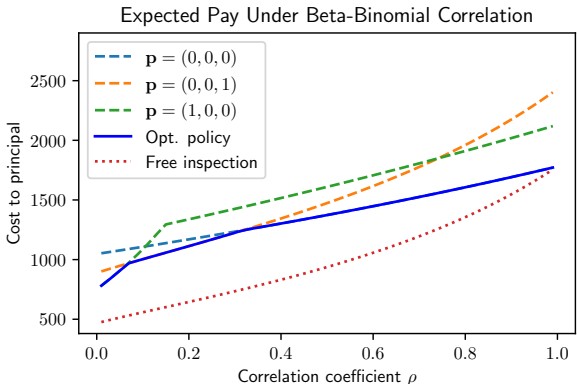
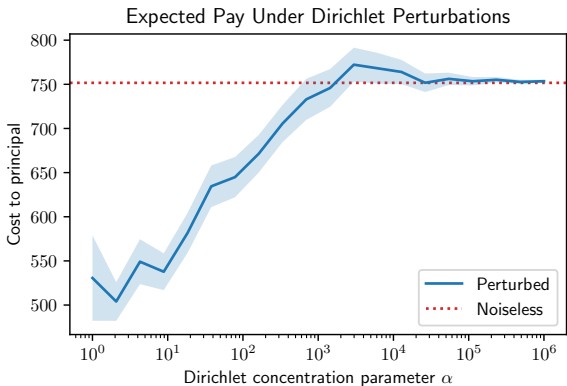

*Figure 9.* Robustness of simple contracts against structured correlation and random perturbations in the SWE-Bench setting (Appendix H.3.2). **(Left)** Optimal contract pay for different levels of Beta-Binomial correlation. Results show that optimal inspection policies remain simple even when parametric correlation is added to a setting satisfying MLRP and ISOP. **(Right)** Optimal pay of Dirichlet-perturbed contracts, showing convergence towards the noiseless setting with increasing concentration.

To evaluate the robustness of simplicity against parametric correlation, we vary the correlation parameter $\rho$, construct the corresponding design problems based on the original costs and Equations (13) and (14), and compute the optimal inspection policy. Results are illustrated in Figure 9 (Left), and show that optimal inspection policies remain simple even when beta-binomial correlation is added. We also note that this property seems to persist across all parameter configurations we sampled empirically, however proving this formally remains an intriguing question for future work. Moreover, results also show that the optimal contract cost increases correlation $\rho$ grows, converging towards the free-inspection baseline $(d_k = 0)$. This suggests that the economic value of additional observations diminishes as their information value decreases.

**Dirichlet perturbations.** To further test the robustness of our findings, we evaluate the model under random Dirichlet perturbations. Given the original distributions $\mathbf{q}^0, \mathbf{q}^1$, and concentration parameter $\alpha > 0$, let:

$$\tilde{\mathbf{q}}_i^0 \sim \text{Dirichlet}\left(\alpha\mathbf{q}_i^0\right) \qquad \tilde{\mathbf{q}}_i^k \sim \text{Dirichlet}\left(\alpha\mathbf{q}_i^1\right) \tag{15}$$

In this setup, $\alpha$ controls the inverse of the perturbation strength: as $\alpha$ increases, the sampled distributions concentrate more tightly around the original distributions. To measure robustness, we vary $\alpha$, sample distributions as described above, and compute optimal adaptive contracts. Results are illustrated in Figure 9 (Right), and confidence bands represent 95% t-test mean confidence intervals. For all repetitions and values of $\alpha$, we observe that the optimal policies are simple. Additionally, we observe that contract cost generally decreases as Dirichlet noise increases. We hypothesize that noise increases the effective distance between the distributions, making them easier to distinguish, and thus decreasing information rent.

**Agent utility and contract cost breakdown.** In Section 6.2, we quantify the trade-off between informativeness and inspection cost, and in Appendix C we describe the possible implications of adaptive inspection on the agent's utility. In this additional experiment, we provide further empirical interpretation. In this experiment, we fix the target action as the top-performing model in our dataset (action $n$), and vary the size $m$ of the refined test suite. We measure the different cost components of the contract: Target action internal cost $c_n$, agent's utility $U_A(n) = T_n - c_n$ (information rent), expected inspection cost $D_n$, and the total cost to the principal $D_n + T_n$ (see Section 2 for notation definitions). In Figure 10 (Left), we observe that the agent's utility generally decreases as the size of the outcome space increases, with a discrete jump in utility as the optimal inspection policy shifts from $\mathbf{p}^* = (0,0,1)$ to $\mathbf{p}^* = (1,0,0)$. We interpret this is an expected decrease in information rent as the signal becomes more informative (see Appendix C for additional discussion). Finally, in Figure 10 (Right) we visualize the cost breakdown of different contract components as $m$ varies, demonstrating the trade-off between information rent and inspection cost as the size of outcome space increases.

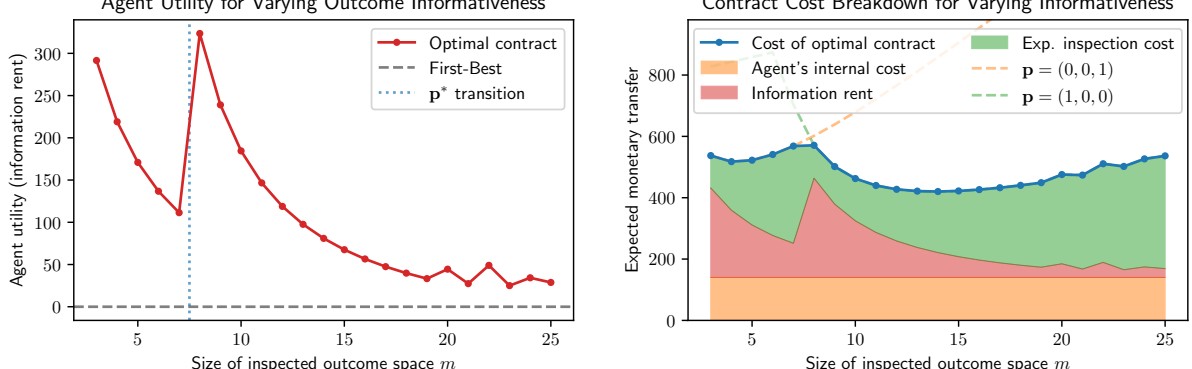

*Figure 10.* Agent's utility and contract cost breakdown as a function of outcome space size (Appendix H.3.2). **(Left)** Agent's utility for the optimal contract. Utility generally decreases as the inspected outcome space becomes more informative, converging towards zero as the information gap narrows. The discrete jump in utility corresponds to a shift in optimal inspection policy. **(Right)** Cost breakdown of the optimal contract, illustrating the trade-off between information rent and inspection cost as the informativeness of the outcome space varies.

