# OpenReview forum: "Adaptive Contracts for Cost-Effective AI Delegation"
_ICML.cc/2026/Conference — ICML 2026 regular_

### Official Review · Reviewer_u2Gf · 2026-03-11

**Soundness:** 3
**Presentation:** 3
**Significance:** 3
**Originality:** 3
**Overall Recommendation:** 5
**Confidence:** 3

**Summary:**

The article explores the use of adaptive contracts to incentivize AI model providers to honestly offer services using appropriate and high-quality models, while optimizing users' utility (reward - payment - inspect cost). It proposes a novel two-stage adaptive contract model combining coarse-grained evaluation and fine-grained inspection. The article analyzes how to solve for the optimal contract under some mild assumptions (constant actions/ISOP), as well as the computational complexity of the solution in general cases. The article also examines the impact of incorporating randomization in contract design on the existence of equilibrium and users' payments. It proposes two model relaxations: uncommitted inspection probability and negative inspection, to ensure the existence of equilibrium under nondeterministic inspection, and compares the optimal payments under various model settings (Fig. 2). Finally, the article validates the improvement of adaptive contracts on user utility and the characterization of optimal contracts through two real-world tasks: QA and agentic coding.

**Compliance With Llm Reviewing Policy:**

Affirmed.

**Final Justification:**

The authors have addressed my concerns in the rebuttal, I therefore maintain my original positive judgement.

**Key Questions For Authors:**

1. The current AI pricing and service delivery strategies are primarily provided by model providers (agents). Can we also consider the changes in agent utility (through reasonable modeling) under adaptive contracts and other contract types?
2. In section 6.1, it is stated that the cost of the model is GPT-4o > GPT-3.5 > GPT-4o-mini. Does this mean that $q^{len<250}$ and $q^{len>=250}$ also violate MLRP? Additionally, based on Appendix G.2 line 1514, the $q^{len<250}$ and $q^{len>=250}$ on Fig 4 (Top) seem to be plotted inversely.
3. In Fig. 5, why doesn't the optimal contract inspect in all cases when the inspection cost d=0, i.e., $p=(1,1,1)$, but only inspects when both tests are successful? Is this to incentivize the agent to choose actions (models) that are more beneficial to the principal's utility?

**Limitations:**

Yes

**Strengths And Weaknesses:**

### Strengths
- Soundness: The theoretical analysis is rigorous and comprehensive, encompassing both possible results and hardness results. Reasonable model relaxations are proposed for encountered theoretical issues (such as the nonexistence of Stackelberg equilibrium under nondeterministic inspection). I have not examined the proof details, but some of the proof ideas are intuitive.
- Presentation: The presentation is clear, and the examples provided and the diagrams used to explain concepts and results greatly assist readers in understanding.
- Signification: The research question focuses on improving the utility of users in AI delegation, enabling users to access high-quality models and reducing their evaluation costs. This has practical significance and is supported by relevant literature.
- Originality: The proposal of adaptive contracts is innovative.
---
### Weakness
See questions

---

> ### Author Rebuttal · Authors · 2026-03-31
>
> Thank you for the encouraging and thorough review! We address your questions below:
>
> > The current AI pricing and service delivery strategies are primarily provided by model providers (agents). Can we also consider the changes in agent utility (through reasonable modeling) under adaptive contracts and other contract types?
>
> After additional empirical analysis and theoretical investigation, we found that the introduction of adaptivity can be either beneficial or detrimental to the agent, depending on the problem parameters.
> In particular, we have identified two scenarios leading to opposite effects on the agent's utility:
>
> * Inspection $\rightarrow$ agent utility decrease: When inspection is affordable and reveals the model chosen by the agent (e.g., when outcome distributions of different models have negligible overlap in support, and $d$ is sufficiently small), then the principal completely eliminates moral hazard by inspecting. In this extreme case, the agent's utility decreases to 0 (i.e. the "first-best" solution). Empirically, in our code-generation experiment (Section 6.2) we observe a decrease in agent utility as the size of the refined test suite grows.
> * Inspection $\rightarrow$ agent utility increase: When a high-cost LLM is not cost-effective for the principal without inspection but becomes reward-maximizing with inspection, the agent's utility may increase due to higher information rent. Empirically, we observe this in our AlpacaEval experiment (Section 6.1), where the most expensive model (GPT-4o) only becomes reward maximizing when inspection is allowed.
>
> The second case is particularly interesting, as it can potentially provide additional motivation for adoption of adaptive contracts in practical settings. We are very grateful for this perspective, and will add the new analysis to the paper.
>
>
> > In section 6.1, it is stated that the cost of the model is GPT-4o > GPT-3.5 > GPT-4o-mini. Does this mean that $q^{len<250}$ and $q^{len>=250}$ also violate MLRP?
>
> Yes. While the contract design setting in Section 6.2 satisfies MLRP and ISOP, the setting in Section 6.1 has a signal distribution which violates MLRP. Our goal in this experiment is to demonstrate the functional form and utility of adaptive of contracts in a setting which goes beyond these theoretical assumptions.
>
> > Additionally, based on Appendix G.2 line 1514, the $q^{len<250}$ and $q^{len>=250}$ on Fig 4 (Top) seem to be plotted inversely.
>
> Thanks! Fixed the notation.
>
> > In Fig. 5, why doesn't the optimal contract inspect in all cases when the inspection cost d=0, i.e., $p=(1,1,1)$, but only inspects when both tests are successful? Is this to incentivize the agent to choose actions (models) that are more beneficial to the principal's utility?
>
> This is an excellent question! In this contract design setting, the two inspections policies that you mention are in fact *both optimal* under zero inspection cost ($d=0$) - A property which is also supported by our theoretical findings.
>
> To explain this using our theoretical framework, note that the contract design setting in Section 6.2 satisfies MLRP+ISOP, and therefore by Theorem 3.5 the optimal adaptive contract  pays for at most one signal and one outcome. Then, by Proposition D.1, there always exists an optimal adaptive contract which inspects one signal at most. This implies that there exists an optimal inspection policy $p$ with support of size at most $1$ for all $d \ge 0$. And indeed, for $d=0$ we obtain that $p=(0,0,1)$ and $p=(1,1,1)$ are both optimal, and $p=(0,0,1)$ is the only optimal policy for strictly positive values of $d$ as $d \to 0$. We will further emphasize this in the paper, and also add a larger graph to the appendix showing the expected utility of all inspection policies (including $p=(1,1,1)$) as a function of $d$ to better clarify.
>
> --
>
> And finally, if any questions remain or arise, please let us know!

---

> > ### Author Rebuttal · Reviewer_u2Gf · 2026-04-01
> >
> > - Regarding Q1, the author illustrates the possible positive and negative impacts of adaptive contracts on agent utility through supplementary experiments. The author provides intuitive explanations for the reasons behind these two scenarios.
> > - Regarding Q2 and Q3, the author's response has addressed my concerns.
> >
> > The overall recommendation is already a 5, so I maintain the score.

---

> > > ### Author Response · Authors · 2026-04-07
> > >
> > > Thank you again for the encouraging feedback, and for the very helpful suggestions! We will gladly incorporate the new results and insights into the revised paper.

---

### Official Review · Reviewer_Pqkh · 2026-03-12

**Soundness:** 3
**Presentation:** 3
**Significance:** 2
**Originality:** 3
**Overall Recommendation:** 3
**Confidence:** 3

**Summary:**

The paper studies contract design when performance evaluation itself is costly and noisy, with AI services as a typical example. Instead of evaluating the AI output once and paying based on that result (as in the standard contract design model), the paper allows the principal to adaptively decide whether to run a more expensive evaluation after seeing a cheap preliminary signal. It then studies how to design optimal payment and inspection policies, and analyzes their computational complexity and economic properties.

**Compliance With Llm Reviewing Policy:**

Affirmed.

**Final Justification:**

I appreciate the authors' responses, which have partially addressed my questions in their rebuttal. I have no objection to the theoretical nature of the paper or to the limited empirical evidence in its current form. My main concern, however, remains whether the proposed model and methodology are truly appropriate for the applications that motivate the work. Given this concern, I would prefer to maintain my original assessment.

**Key Questions For Authors:**

Can you respond to the modeling weaknesses pointed out above?

**Limitations:**

Yes.

**Strengths And Weaknesses:**

The paper is this a clean extension of standard contract design to settings with adaptive information acquisition. The framework is presented clearly and captures the idea that monitoring quality may itself be costly and sequential. The authors make an effort to analyze the computational aspects of the model, providing algorithmic results for special cases and hardness results for the general problem.

That said, I am less convinced about the practical significance of the model. Several assumptions appear too strong to meaningfully capture real AI deployment settings:

- The paper assumes that the principal knows the distributions of evaluation outcomes for each action. This seems unlikely in realistic AI settings where model behavior and benchmarks evolve very quickly. It also assumes a relatively small discrete action space for the agent. The provider (agent) chooses among a few models with known costs, whereas in practice providers can adjust behavior along many dimensions and perhaps use multiple models in a task.

- The motivating moral-hazard scenario, where providers secretly substituting cheaper models, seems somewhat misaligned with current LLM markets, which rely more on posted pricing and reputation than on bespoke contracts.

- The model compresses evaluation into a small discrete signal/outcome space and a single optional inspection step. This might be an oversimplification of the multidimensional and iterative real evaluation pipelines. (As a minor point, the terminology distinguishing the first-stage evaluation as a "signal" and the second-stage evaluation as an "outcome" is somewhat confusing, since both are simply evaluation results revealed at different stages.)

Hence, overall, while the paper presents an interesting theoretical extension of contract design and provides solid algorithmic analysis, the practical significance appears quite limited. Since the paper is motivated primarily by this specific application scenario, I am less convinced that the work stands strongly when it consists of mainly theoretical results that are somewhat removed from practical applicability.

---

> ### Author Rebuttal · Authors · 2026-03-31
>
> Thank you for the thoughtful review. We address your points below:
>
> > The paper assumes that the principal knows the distributions of evaluation outcomes for each action. This seems unlikely in realistic AI settings where model behavior and benchmarks evolve very quickly.
>
> The optimization program we present indeed assumes that the score distributions are known by the principal and remain fixed for the duration of the contract.
> In response to similar concerns about the strength of these assumptions, the existing contract design literature (without adaptive inspection) offers methods for *learning optimal contracts* in dynamic environments that may evolve over time (e.g., [1,2]),
> and for *robust contract design*, which often only requires partial and easier-to-track knowledge about the prior distribution (e.g., knowledge of expected values instead of full distributions [3]).
> Both of these approaches typically use classic contracts as a basic building block or benchmark, and we hope that our results will motivate their extension to adaptive contract design settings. We will emphasize this point in our discussion.
>
>
> > It also assumes a relatively small discrete action space for the agent. The provider (agent) chooses among a few models with known costs, whereas in practice providers can adjust behavior along many dimensions and perhaps use multiple models in a task.
>
> While our theoretical framework only assumes that the space of possible model configurations is finite, the size of this space indeed plays a very important role in the context of practical deployment. Similar to the answer above, in response to such concerns there is a rich body of work on non-adaptive contract design with *combinatorial actions* [4]. Generally, the combinatorial approach recovers tractability guarantees by making additional assumptions on the structure of the agent's action space (e.g., assuming that adding different run-time optimizations to the LLM inference process has specific "diminishing marginal return" properties). While our aim in this work is to keep assumptions minimal, analyzing adaptive contracts in a combinatorial action setting is a natural research direction.
>
> > The motivating moral-hazard scenario, where providers secretly substituting cheaper models, seems somewhat misaligned with current LLM markets, which rely more on posted pricing and reputation than on bespoke contracts.
>
> While adoption of AI by individuals is increasing, a very significant proportion of the AI services market is dominated by business and enterprise clients. For example, Reuters has recently reported that ~80% of Anthropic’s revenue comes from business clients [5], and that ~30% of OpenAI's revenue comes from enterprise offerings [6].
>
> Enterprises and large businesses are typically able to benefit from the adaptive contract design framework we propose, as they tend to have bargaining power which allows them to negotiate pricing. In addition, business clients typically commit to long-term agreements and have higher vendor lock-in due to the technical integration efforts required, increasing the risks of strategic behavior and "corner-cutting" by AI service providers over time.
>
> > The model compresses evaluation into a small discrete signal/outcome space and a single optional inspection step. This might be an oversimplification of the multidimensional and iterative real evaluation pipelines. (As a minor point, the terminology distinguishing the first-stage evaluation as a "signal" and the second-stage evaluation as an "outcome" is somewhat confusing, since both are simply evaluation results revealed at different stages.)
>
> Given our results in this paper, the extension to multiple inspection steps indeed seems like a very natural question for future work. We use the terms "signal" and "outcome" to make the two stages easier to distinguish within the theoretical analysis, and we also agree that a more "uniform" terminology would perhaps be more natural if the inspection model is generalized to more inspection steps. We will add this point to our discussion.
>
> --
>
> Finally, we hope that our response helps contextualize our contributions. If additional questions arise, we'd be happy to continue the discussion!
>
> References:
> [1] "Adaptive contract design for crowdsourcing markets: bandit algorithms for repeated principal-agent problems". Ho et al. (EC 2014)
> [2] "The sample complexity of online contract design". Zhu et al. (EC 2023)
> [3] "Simple versus Optimal Contracts". Dütting et al. (EC 2019)
> [4] "Combinatorial Contracts". Dütting et al. (FOCS 2021/SICOMP 2025)
> [5] "US tech startup Anthropic unveils cheaper model to widen AI's appeal". Reuters report 2025-10-15.
> [6] "OpenAI’s hype machine faces a corporate challenge". Reuters report 2025-10-29.

---

> > ### Author Rebuttal · Reviewer_Pqkh · 2026-04-03
> >
> > Thank you for the detailed response and clarifications. I appreciate the effort to contextualize the modeling assumptions and point to related literature. While some of my questions have been addressed, I still have reservations about how well the modeling choices capture realistic AI deployment settings and about the practical relevance of the proposed framework. The action spaces in modern AI applications are often much larger and much more unstructured than those considered in combinatorial contract design, and the learning or robust contract design approaches cited generally target small models. As a result, these points do not fully address my concerns. I do recognize and appreciate the effort involved in developing the theoretical results, but overall my assessment remains unchanged.

---

> > > ### Author Response · Authors · 2026-04-07
> > >
> > > Thank you for the feedback! We truly value the perspective illustrated by your remaining reservation about the nature of action spaces in modern AI applications, and will expand our discussion of limitations to reflect this.
> > >
> > > In addition, and without reducing our commitment above, we would also like to emphasize two points that we believe could be helpful additions to the discussion:
> > >
> > > First, while the possible number of model variants is indeed theoretically unlimited, we note that many providers in the AI supply chain effectively only face a limited set of options. For example, within the enterprise sector, recent reports indicate that AI services are becoming increasingly embedded within existing business software [1]. Providers of these software products act as "intermediary" AI providers, which process the initial request and send it to one of the foundation model providers (e.g. OpenAI, Anthropic or Google) via an API call. The set of actions available to such providers is the set of compatible API endpoints, and we believe it is reasonable to assume that this set is finite and reasonable in size: For example, OpenAI currently offers a selection of ~40 text models through its API, and OpenRouter offers a selection of ~600 text models through its cross-market API.
> > >
> > > Second, we would also like to note that the body of literature on learning and robust contract design is large, and some works have significantly relaxed assumptions regarding the cardinality of the action space. In particular, the robust contract design results of Carroll [2] show that it is possible to design incentive-compatible contracts even in settings where the principal only has information about a finite and possibly small subset of an action space that may be infinite.
> > >
> > > Finally, we also acknowledge that the AI inference systems landscape is complex and constantly evolving, and new insights about the structure of cost-benefit trade-offs can facilitate the design of more efficient contracts. With this, we also hope that our work will motivate efforts to further characterize the structure of cost-benefit considerations in modern AI deployments.
> > >
> > >
> > > --
> > >
> > > References:
> > >
> > > [1] "Gartner Predicts 40% of Enterprise Apps Will Feature Task-Specific AI Agents by 2026, Up from Less Than 5% in 2025". Gartner report 2025-08-26.
> > > [2] "Robustness and Linear Contracts". Carroll (American Economic Review 2015).

---

### Official Review · Reviewer_Kuwg · 2026-03-18

**Soundness:** 4
**Presentation:** 4
**Significance:** 4
**Originality:** 4
**Overall Recommendation:** 5
**Confidence:** 2

**Summary:**

This paper studies adaptive contracts for LLM usage, where the user may request a costly second-round evaluation before paying the LLM provider for performance. The author characterizes the computational complexity of calculating the adaptive contract, proposing conditions where the problem is feasible, and proves the difficulty when the conditions are not met. Finally, the author showed how randomization may break the equilibrium and how equilibrium can be established by exerting restrictions on the users.

**Compliance With Llm Reviewing Policy:**

Affirmed.

**Final Justification:**

The rebuttal has addressed my concern. I hold my assessment that this paper is overall valuable contribution to the community. My weights on the strength and weakness hasn't changed much by the rebuttal. I hold a positive opinion on acceptence.

**Key Questions For Authors:**

* This is a very interesting and timely problem, and the proposed contract differs from the current charge-by-token schema. This model assumes the provider takes the resources and efforts for evaluation. In practice, different customers may have different needs. So, will the conclusions here change much if the principal takes its own initiative for the inspection?
* Does the proposed adaptive contract apply to models (e.g., object detection, person identification, etc.) other than LLM? In other words, the length of tokens in the inputs and responses does not seem to be factored into this contract model.

**Limitations:**

Yes

**Strengths And Weaknesses:**

The paper is technically sound. The problem is well analyzed from sufficient and necessary directions. For the hardness issue, the authors also provide conditions that are both practical and reasonable.

The presentation is clear, and the problem is well motivated. The findings are significant in that this paper addresses a timely problem of pricing LLM services, which big AI companies have been facing for years.

The problem is overall well studied. However, a few points may need further improvement:
* Missing literature comparison on the result side: How did existing works address the contract computation problem in their algorithms?
* The model assumes the agent's action space is choosing from a set of models. In reality, with moral hazard, it is likely that the agent emits signals or games the evaluation to maximize its own profit, a phenomenon not captured by the model.

---

> ### Author Rebuttal · Authors · 2026-03-31
>
> Thank you for the encouraging review and insightful comments! We address your questions below:
>
> >  Missing literature comparison on the result side: How did existing works address the contract computation problem in their algorithms?
>
> Without adaptivity, optimal contracts can be computed in time polynomial in the number of actions and the number of outcomes. This is due to the fact that the non-adaptive contract optimization task can be formulated as a linear program
> (e.g., [1], Section 3). We will be sure to note this in Section 3 in our final version.
>
> > The model assumes the agent's action space is choosing from a set of models. In reality, with moral hazard, it is likely that the agent emits signals or games the evaluation to maximize its own profit, a phenomenon not captured by the model.
>
> This is a very good point. While naive evaluation criteria can indeed be manipulated (e.g., biasing the model towards longer or shorter answers), we note that many evaluation methods of significant practical interest are in fact quite robust to gaming. For example, in code generation (discussed Example 1.1 and Section 6.2), it is generally very challenging to pass unit tests without generating the correct program. As additional examples, direct evaluation by humans (as discussed in AlpacaEval [2]) and LLM evaluation based on forecasting (as discussed in [3]) are both generally quite robust to this type of gaming.
>
> The adaptive contracts framework supports the full spectrum of evaluation methods, regardless of their susceptibility to manipulation. However, we also agree that this type of robustness is a significant aspect in practical deployment, and we hope that this work will further motivate the consideration of strategic elements in the design of evaluation methods. We will further emphasize and discuss this point in the paper.
>
> > This is a very interesting and timely problem, and the proposed contract differs from the current charge-by-token schema. This model assumes the provider takes the resources and efforts for evaluation. In practice, different customers may have different needs. So, will the conclusions here change much if the principal takes its own initiative for the inspection?
>
> If we understand your question correctly, you are right to point out that our model considers a contract between a single customer and a provider, not a principal representing a heterogeneous group of customers. We
> didn't consider such extensions, but it seems like some notions of "collective contracting" could still fall under our model, by viewing a random sample of a customer from the pool of customers as part of the randomness involved in mapping actions to signals and outcomes.
>
> > Does the proposed adaptive contract apply to models (e.g., object detection, person identification, etc.) other than LLM? In other words, the length of tokens in the inputs and responses does not seem to be factored into this contract model.
>
> Definitely. While we focus on LLM text generation as our central motivating application, our model and results rely on general assumptions which are likely to apply more broadly. In particular, our tractability results will apply to any generative AI/computer vision application with an evaluation method that satisfies ISOP+MLRP (or with constant-many actions/signals).
> Our paper contributes to a broader research program within algorithmic contract design investigating structural properties that guarantee the tractability or simplicity of optimal contracts.
>
> --
>
> Finally, if any questions remain or arise, please let us know!
>
> References:
> [1] "Algorithmic Contract Theory: A Survey". Dütting et al. (2024)
> [2] "AlpacaEval: An Automatic Evaluator of Instruction-following Models". Li et al. (2023)
> [3] "LLM-as-a-Prophet: Understanding Predictive Intelligence with Prophet Arena". Yang et al. (ICLR 2025)

---

> > ### Author Rebuttal · Reviewer_Kuwg · 2026-04-04
> >
> > I appreciate the authors' detailed response. The rebuttal has resolved all my concerns. In my opinion, this paper has achieved great progress in a difficult but urgent problem. I would maintain my score.

---

> > > ### Author Response · Authors · 2026-04-07
> > >
> > > Thank you for the feedback! We are very grateful for the discussion, and will incorporate the new insights into the revised paper.

---

### Decision · Program_Chairs · 2026-04-30

**Decision:**

Accept (regular)

**Comment:**

The reviewers favorably evaluated the significance of the proposed model and its claims, the rigor of the analysis, and the clarity of the exposition, while some reviewers have concerns about the practicality of the approach given the gap between the model and real-world problems. In particular, concerns remain about whether the proposed model is genuinely motivated by its target applications and whether it is likely to be useful in the long term.

However, while the concerns about practicality are an important perspective, this paper is primarily focused on theory, and from a long-term viewpoint, the issue of direct practical applicability does not fully undermine the paper's value. In addition, given that the majority of reviewers are supportive of the paper overall, I recommend acceptance of this paper.

I strongly recommend that the authors revise the paper to add discussion of whether the proposed model is well motivated by the real-world problems it targets, and of the potential long-term usefulness of the obtained results.